# Using recurrent neural network to estimate irreducible stochasticity in human choice behavior

Yoav Ger[1]*, Moni Shahar[2], Nitzan Shahar[1,3]

[1]Sagol School of Neuroscience, Tel Aviv University, Tel Aviv, Israel; [2]TAD, Center of AI & Data Science, Tel Aviv University, Tel Aviv, Israel; [3]School of Psychological Sciences, Tel Aviv University, Tel Aviv, Israel

## eLife assessment

In this study, Ger and colleagues present a **valuable** new technique that uses recurrent neural networks to distinguish between model misspecification and behavioral stochasticity when interpreting cognitive-behavioral model fits. Simulations provide **solid** evidence for the validity of this technique and broadly support the claims of the article, although more work is needed to understand its applicability to real behavioral experiments. This technique addresses a long-standing problem that is likely to be of interest to researchers pushing the limits of cognitive computational modeling.

*For correspondence:
yoavger@mail.tau.ac.il

Competing interest: The authors declare that no competing interests exist.

**Abstract** Theoretical computational models are widely used to describe latent cognitive processes. However, these models do not equally explain data across participants, with some individuals showing a bigger predictive gap than others. In the current study, we examined the use of theory-independent models, specifically recurrent neural networks (RNNs), to classify the source of a predictive gap in the observed data of a single individual. This approach aims to identify whether the low predictability of behavioral data is mainly due to noisy decision-making or misspecification of the theoretical model. First, we used computer simulation in the context of reinforcement learning to demonstrate that RNNs can be used to identify model misspecification in simulated agents with varying degrees of behavioral noise. Specifically, both prediction performance and the number of RNN training epochs (i.e., the point of early stopping) can be used to estimate the amount of stochasticity in the data. Second, we applied our approach to an empirical dataset where the actions of low IQ participants, compared with high IQ participants, showed lower predictability by a well-known theoretical model (i.e., Daw's hybrid model for the two-step task). Both the predictive gap and the point of early stopping of the RNN suggested that model misspecification is similar across individuals. This led us to a provisional conclusion that low IQ subjects are mostly noisier compared to their high IQ peers, rather than being more misspecified by the theoretical model. We discuss the implications and limitations of this approach, considering the growing literature in both theoretical and data-driven computational modeling in decision-making science.

## Introduction

Humans' behavior is thought to arise from a set of multidimensional, complex, and latent cognitive processes. Theoretical computational models allow researchers to handle this complexity by putting forward a set of mathematical descriptions that map assumed latent cognitive processes to observed behavior (*Daw, 2011*; *Smith and Ratcliff, 2004*). By fitting a theoretical model (also sometimes termed

'symbolic' models) to the observed behavior, the researcher is able to draw conclusions regarding the estimation and architecture of the latent cognitive processes (*Wilson and Collins, 2019*). While this approach has already led to substantial scientific findings (*Montague et al., 2012*; *Rescorla, 1972*), it still faces a major challenge since the true underlying model is always left unknown (*Box, 1979*). Specifically, theoretical computational models usually stem from strict theoretical assumptions that can differ from the true cognitive mechanism that led to the observed behavior, a problem termed 'model misspecifications' (*Beck et al., 2012*; *Nassar and Frank, 2016*). Moreover, the issue of model misspecification is even more critical when considering that most studies in the field focus on group differences and therefore might neglect individual variation in the expressed models (*Stephan et al., 2009*; *Rigoux et al., 2014*). In the current study, we address the fundamental methodological problem of theoretical model misspecification by using well-established theory-independent models, a family of highly flexible models that require no prior theoretical specification.

Imagine that you are working on a theoretical question that you have addressed well by assembling a computational model that, after much effort, can replicate and mimic empirical observations. Still, some data will not be accurately predicted by the model, leaving an open question regarding the possibility that better specification of the model (i.e., changing some mechanisms, adding additional processes) would better capture the pattern in your observations. Currently, to address the issue of model misspecification, methodology in theoretical computational science calls for researchers to perform a model comparison analysis (*Wilson and Collins, 2019*). Here, the researcher is encouraged to put forward a set of different candidate models and quantify which model is most plausible given the observed data (*Palminteri et al., 2017*). Yet, even after a rigorous process of model comparison, the best-fitting model will still have what seems like room for improvement. This leaves an unanswered question regarding the possibility that yet another different model that was not described by the researcher might better explain and predict the observations (*Eckstein et al., 2021*; *McElreath, 2020*).

The distance in terms of variance between observed and predicted model behavior is typically termed the predictive gap. The predictive gap between individuals' behavior and model prediction can be attributed to two factors: The first is model misspecification, as mentioned above. Here, the individual's true behavior-generating model is different from the one suggested by the researcher (*Beck et al., 2012*; *Nassar and Frank, 2016*). The second factor is stochasticity, which refers to true noise or the natural randomness in human behavior (*Faisal et al., 2008*; *Findling et al., 2019*). The underlying assumption is that some variance in behavior is unpredictable and therefore represents irreducible variance. The assumption of irreducible variance is well accepted across scientific disciplines (*Griffiths and Schroeter, 2018*), and it suggests that even Laplace's demon (*Gleick, 2011*), a metaphorical demon that is assumed to have access to all mechanisms and processes in the universe, would not be able to produce perfect predictions. Both model misspecification and irreducible noise influence the predictive gap, yet identifying the contribution of each factor may lead to different implications. A predictive gap attributed mostly to model misspecification suggests that the researcher needs to increase the space of candidate models to further reduce the gap. However, a predictive gap attributed mostly to irreducible noise suggests that such improvement is mostly out of reach.

More recently, data-driven approaches based on neural networks have emerged as an alternative modeling paradigm in cognitive research (*Dezfouli et al., 2019b*; *Song et al., 2021*). These networks, which are models trained to learn directly from data rather than relying on theoretical assumptions about human behavior, have been shown to surpass typical theoretical models in pure action prediction (*Dezfouli et al., 2019b*; *Song et al., 2021*). The key property of these models is that a priori, the models' parameters do not directly map to some underlying cognitive process. Instead, the free parameters (i.e., weights) are iteratively adjusted during training to improve the network's predictive capability for a desired objective function (*LeCun et al., 2015*). Furthermore, unlike classical theoretical models, neural networks are overparameterized models that include a large number of learnable free parameters. This property allows the network high flexibility and the ability to approximate a wide range of functions (*Siegelmann and Sontag, 1992*; *Hornik et al., 1989*), including functions believed to arise from human cognition (*Barak, 2017*). For example, *Dezfouli et al., 2019b* trained a recurrent neural network (RNN) to predict future human actions in a two-armed bandit task. Their study indicated that the RNN is capable of surpassing baseline RL models in action prediction. In another work, *Fintz et al., 2022* trained an RNN model to predict future human actions in a four-armed bandit task.

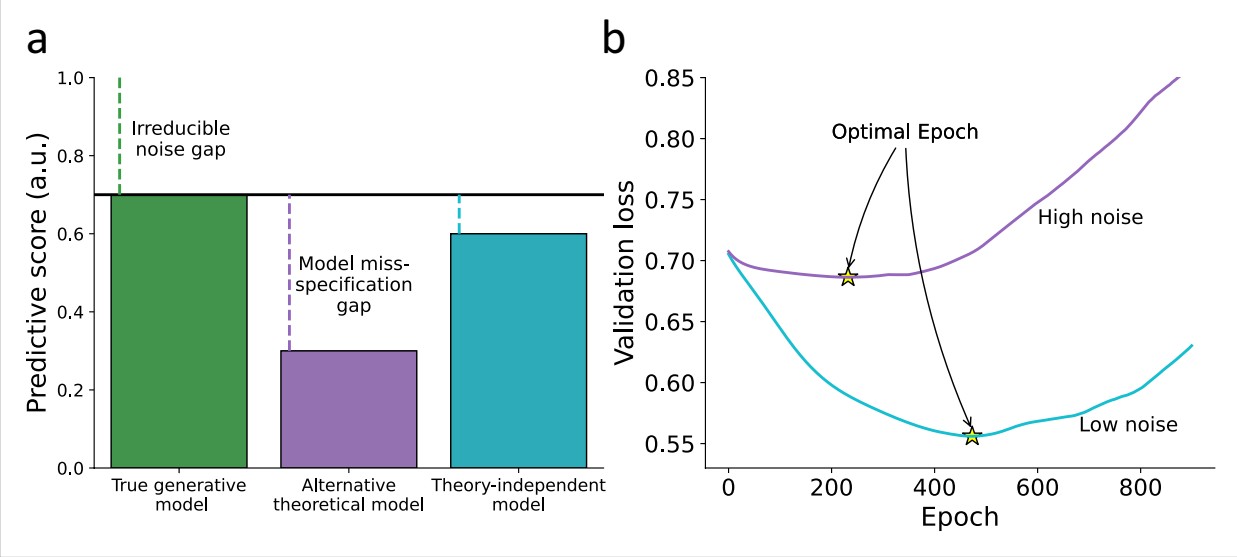

**Figure 1.** Hypothetical illustration for using theory-independent models to explore the fit of theoretical models to human behavior. (**a**) Predictive performance – we illustrate the predictive gap for three hypothetical models: first, the true data-generating model (i.e., forever unknown to the researcher; green) where the remaining gap is due to an irreducible noise component in the individual's behavior. Second, a hypothetical alternative theoretical model (e.g., specified by a researcher; purple), where the remaining predictive gap reflects both irreducible noise and model misspecification. Finally, a hypothetical theory-independent model (turquoise) is assumed to reflect a predictive gap that is mainly due to model misspecification compared with the alternative model, yet does not provide a clear theoretical interpretation. We, therefore, assume that theory-independent models can be used to inform researchers about the amount of improvement that can be further gained by assembling additional theoretical models. (**b**) Point of early stopping – when training a network, we can examine its performance against a validation set as a function of the number of epochs used for training the parameters (x-axis). The point of early stopping reflects the maximum number of training epochs with the best predictive performance, just before the network starts to overfit the data (i.e., learn patterns that are due to noise; indicated by a yellow star). Here, we illustrate two hypothetical learning curves reflecting the point of early stopping for low-noise (purple line) and high-noise (blue line) datasets. Specifically, we illustrate the notion that the point of early stopping can reflect the amount of noise in the data, so a lower point of early stopping reflects noisier data (considering a fixed number of observations for the two datasets and network size).

They showed that the RNN was able to capture atypical behaviors such as choice alternations and win-shift-lose-stay, which common cognitive models failed to capture. Finally, *Song et al., 2021* trained an RNN model to predict the choices of humans in a reinforcement learning (RL) task that included a rich space of possible states and actions. They showed that the RNN was able to outperform the best-known cognitive model.

However, the high flexibility of neural networks also comes with the disadvantage that these networks are often considered black box models. Despite attempts to interpret the networks' latent space (*Dezfouli et al., 2019a*), it is still not clear how to efficiently interpret behavior using them, which is a major goal in cognitive research (*Hasson et al., 2020*; *Yarkoni and Westfall, 2017*). In this study, we aim to leverage the network's high flexibility and predictive capability to address the problem of identifying misspecification in theoretical computational models, using two different estimates.

## Predictive performance

We assume that, for a fixed dataset, the flexibility of neural networks would enable them to capture the mapping between independent variables and dependent variables, up to a point where the remaining predictive gap is primarily due to noise rather than model misspecification (see *Figure 1a*). Therefore, we hypothesize that across different true generative models, neural networks will reach a predictive gap that closely resembles the predictive gap that remains when the true generative model is known. As a result, theory-independent models can serve as an upper benchmark against which the new data can be predicted based on previous observations. If, for some individuals, the theoretical model is severely misspecified, we expect to observe a significant improvement in our ability to predict unseen data using theory-independent models. However, if an individual is simply exhibiting a noisier pattern of behavior, we anticipate little to no improvement when using theory-independent models.

## Point of early stopping

Neural networks are prone to overfitting, where they fit too closely to the training data and fail to generalize to new datasets (*LeCun et al., 2015*). To mitigate overfitting, a common technique called early stopping is used (*Bishop and Nasrabadi, 2006*). In early stopping, the network is trained for multiple epochs, and the predictive performance on a validation set is monitored. Initially, the performance improves as the number of epochs increases, but there comes a point where the network starts learning patterns that reflect noise rather than true underlying patterns in the data. The point of early stopping is determined as the epoch at which the best prediction is achieved. We propose that the point of early stopping largely reflects the amount of noise in the data. With a fixed dataset and network size, an earlier stopping point indicates noisier data (see *Figure 1b*). The rationale behind this hypothesis is that, under certain realistic assumptions, the probability of stopping at an earlier epoch is higher for a sequence with more noise. This is because the noisy sequence has a lower ratio of true signal to noise, resulting in less information learned by the algorithm over the course of training. Using the point of early stopping to estimate irreducible noise has advantages, including the potential to require fewer data compared to using network predictive accuracy to estimate model misspecification. Estimating the upper bound of the prediction accuracy of the true data generation model using predictive accuracy typically necessitates three sets of data per individual (training, validation, and test). This approach may also require initial auxiliary data for pre-training the recurrent neural network (RNN) to achieve optimal results. However, estimating the point of early stopping can be accomplished with only two sets of data per individual (training and validation). Additionally, this approach may benefit from initializing the network with random weights instead of a pre-trained RNN, as it allows for estimating the maximum possible variance in the optimal epoch estimate among individuals.

In the present study, our aim is to estimate model misspecification in individual participants. We propose a method that utilizes theory-independent models, specifically RNNs, which possess high flexibility in learning complex features from data without manual engineering. In Study 1, we conducted simulations involving three groups, each consisting of $N = 100$ simulated agents performing a two-step RL decision task. Each group of agents followed a specific data-generating model, and the agents differed in the level of true noise present in their actions. We assumed three hypothetical labs, with each lab having knowledge of only one data-generating model, thus misspecifying two-thirds of the agents. We demonstrated that the predictive performance of a pre-trained RNN using three datasets per agent (training, validation, test) could be used to classify whether an agent was misspecified by the lab's theoretical model. Additionally, we showed that the number of optimal training epochs for an RNN with random weights and fewer data (training and validation, without a test set) could serve as an estimate for comparing the amount of noise in agent data, given a fixed dataset and RNN architecture. Next, in Study 2, we analyzed an empirical dataset with $N = 54$ participants who completed three sessions of a two-step decision task in a lab setting. We examined the fit of a well-known theoretical model (Daw's hybrid model) to participants with different IQ levels. We found that the theoretical model showed a systematically poorer fit to participants with low IQ compared to those with high IQ. By utilizing RNN predictive performance and optimal epochs estimates, we classified the source of the low theoretical model fit in low IQ participants. Our findings converged to suggest that the percentage of model miss-specification did not differ between low and high IQ individuals. Instead, low IQ individuals exhibited noisier decision-making compared to their high IQ counterparts. We propose that theory-independent models, such as RNNs, can be valuable in classifying model misspecification. However, we acknowledge the limitations of our approach and discuss potential directions for future research.

## Results

### Study 1: Simulation study

We begin by applying our method to simulated data, where different types of artificial agents governed by distinct generating models performed a two-step task (*Daw et al., 2011*). This simulation allowed us to evaluate our approach using artificial data, where we knew the true underlying data-generating model and the level of true noise for each agent. Our main objective was to assess the ability of theory-independent models to inform us about the factors contributing to a poorer fit

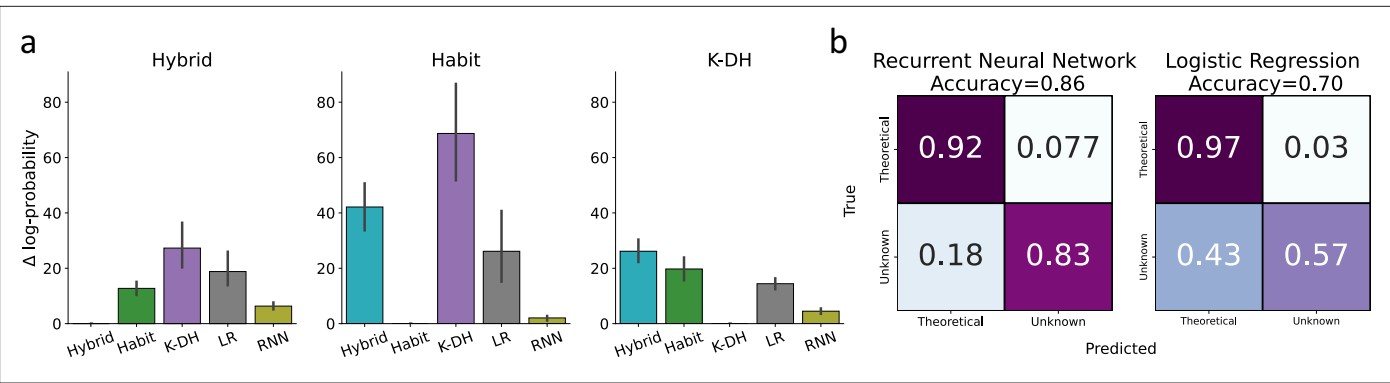

**Figure 2.** Classification of model misspecification using the predictive performance of theory-independent models. Here, we tested our ability to identify model misspecification of theoretical models using recurrent neural network (RNN) and logistic regression. (**a**) Across three theoretical models (Hybrid, Habit, K-DH), we simulated 100 agents and predicted agents' actions using the same three theoretical models, RNN and logistic regression. We present on the y-axis differences in negative log-probability estimates for each fitted model against the true generative model. For example, the left panel depicts the difference in log-probability estimates across all five models for 100 agents simulated using the hybrid model. As expected, across all three generative theoretical models, RNN achieved the best performance score, second only to the true generative theoretical model (black lines represent 95%CI). (**b**) We calculated a confusion matrix for a hypothetical lab that is familiar only with one theoretical model and uses RNN or logistic regression to try and conclude whether a certain agent shows a high predictive gap due to model misspecification. Each cell represents the average value across three classification rounds. We assumed one lab theoretical model as the true data-generating model in each round. Agents were then classified into two classes (assumed lab model or unknown model) based on the difference in predictive scores between the assumed theoretical model and a theory-independent model. For example, the top-left cell in the left confusion matrix indicates the percentage of agents (averaged over three classification rounds) better predicted by their true data-generating model than by the RNN. Results show good classification by RNN (accuracy ≈ 0.86), and logistic regression (accuracy ≈ 0.70), with better performance for RNN compared with logistic regression. Overall, these results suggest that RNN/logistic regression can be used to some extent to inform researchers regarding model misspecification when using theoretical computational modeling.

The online version of this article includes the following figure supplement(s) for figure 2:

**Figure supplement 1.** Action prediction and agent classification for different lengths of simulated trials.

**Figure supplement 2.** Agent classification of each hypothetical lab.

of a theoretical model, specifically model misspecification or irreducible noise. To accomplish this, we introduced three theoretical models (Hybrid model-based/model-free, Habit, and K-dominant hand) and two theory-independent models (logistic regression [LR] and RNN). We generated data from the three theoretical models and fitted all five models to the data from each respective theoretical model. Subsequently, we used the predictive capability of the RNN and LR models, as well as the number of optimal training epochs for the RNN, to estimate and distinguish between model misspecification and stochastic behavior (see 'Materials and methods' for more details regarding the task, theoretical and theory-independent models, and model fitting procedure).

## Classification of model misspecification using predictive performance

Our first aim was to assess the ability of theory-independent models to predict agent choice data across each theoretical model. For each agent in the artificial dataset and for each cross-validation (CV) round, we used one block (training set) to estimate optimal parameters (separately for each model). Then, the optimal parameters were used to predict the agent's first-stage choice data on a withheld block (test set; see 'Materials and methods' for full details). We repeated this procedure for three rounds and averaged the predictive score over all withheld blocks (i.e., $nLP_i^m$; see *Equation 22*). Namely, for each agent we obtained five predictive scores, corresponding to the five models mentioned above (Hybrid, Habit, K-DH, RNN, LR). We found that across all theoretical models, RNN achieved the second-best predictive score, second only to the true generative model (see *Figure 2a*). This suggests that RNN is flexible enough to approximate a wide range of different behavioral models. Moreover, we found that the LR model achieved a poorer predictive score of agent choice data, implying that the model is less expressive compared with RNN.

Next, we performed a hypothetical experiment in which three hypothetical labs estimated individual performance using a theoretical model. Each hypothetical lab was assumed to be aware of only one data-generating model, namely, 'Hybrid-lab', 'Habit-lab', and 'K-DH-lab'. The notion here is that

we illustrate a hypothetical situation where unbeknownst to the research lab, subjects are performing the task under different models. For example, the hypothetical Hybrid-lab is assumed to be aware only of the hybrid model and wrongly assumes that all subjects acted according to the Hybrid model. In such a case, all agents with a true generative model of Habit and K-DH models will be misspecified. However, we assumed that the lab has no knowledge regarding these alternative models, and thus will fit the hybrid model to all agents. We were interested in testing the extent to which theory-independent models (RNN, LR) can help such a lab to classify if a certain agent might be better explained by another unknown model compared to the hybrid model. For this aim, we performed three classification rounds, where at each round we assumed one hypothetical lab classified each agent into one of two classes: lab theoretical model or unknown alternative model. The classification was done based on the difference between the $nLP_i^m$ scores of the assumed theoretical model and the two theory-independent models of each agent. We averaged across all classification rounds and present two confusion matrices, classification by RNN and by LR (see *Figure 2b*). Like before, we found that RNN achieved a higher classification accuracy of 86% compared to the LR model, which reached only 70% accuracy. This finding supports our claim that RNN can signify if a certain subject might be better explained by another unknown model. Importantly, to illustrate the robustness of our results, we provide a supplementary analysis where the exact same analysis was repeated across 100 and 500 observations per agent and found very similar results (see *Figure 2—figure supplement 1*). Furthermore, to assert that this effect is not due to averaging across all classification rounds in the supplementary information, we provide the confusion matrices for each hypothetical lab (Hybrid, Habit, K-DH). We found that across all hypothetical labs the RNN achieved a higher true negative rate and a lower true positive rate compared to the LR model (see *Figure 2—figure supplement 2*).

## Using the number of optimal epochs to estimate noise

When fitting an RNN, we estimate the number of optimal training epochs that minimize both underfitting and overfitting. We reasoned that for a fixed number of observation and network parameters, the point of early stopping (henceforward, 'optimal epochs') should also reflect the amount of noise/information in the behavioral data. Specifically, we hypothesized that the probability of stopping in an earlier epoch is higher for noisier agents since this probability is determined by the ratio between the true signal learned and the noise. To test our hypothesis, we examined the correlations between the number of optimal epochs and the amount of true noise each agent holds (see *Figure 3a*). As expected, we found that agents with high levels of true noise in their data also showed lower number of optimal epochs (i.e., required less RNN parameter training) compared with less noisy agents (see *Figure 3a*; linear correlation coefficient $r = -.67, -.74, -.62$ for the Hybrid, Habit, and K-DH agents, respectively; p<0.001 across all correlation). Importantly, we found that this result holds regardless of the agent's true underlying data-generating model, suggesting that the point of early stopping may be used as an index for the amount of true noise each participant holds. We performed the same analysis with different network sizes and different number of observations (i.e., trials) per agent and found very similar results (see *Figure 3—figure supplement 1*). Therefore, we conclude that the number of optimal epochs reflects the amount of information in the observed behavior. Note that the number of optimal epochs is not an absolute estimate since on its own it can be influenced by other factors, including the number of observations and the size of the net.

## Study 2: Empirical study

We now present an application of our method to an existing dataset, where humans performed an identical two-step task as reported in Study 1, at three different time points (*Kiddle et al., 2018*; see 'Materials and methods'). We predict participants' behavior using a well-established hybrid model (see 'Hybrid model') and demonstrate that low IQ is associated with a higher predictive gap. We put our method to use and examine whether action prediction of RNN can improve the predictive gap of the theoretical model.

Specifically, we assume that if low IQ participants are more frequently misspecified by the hybrid model, then RNN will show a greater reduction of the predictive gap for low compared with high IQ individuals (see *Figure 4a*). However, another possibility is that the higher predictive gap of low compared with high IQ participants in the theoretical hybrid model is mostly due to the noisier

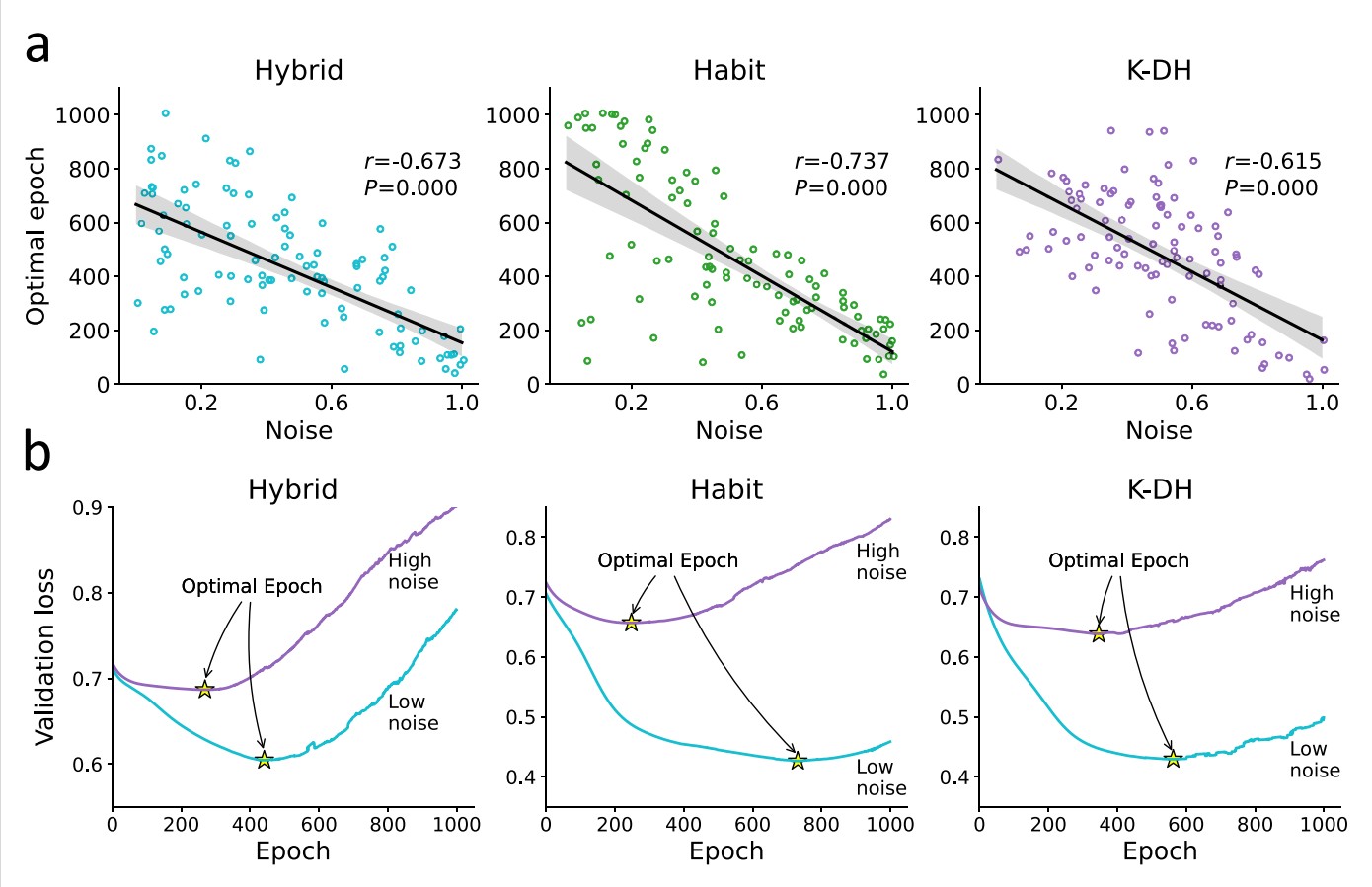

**Figure 3.** Optimal epoch relation to true noise. For each agent, we iteratively trained the recurrent neural network (RNN) while examining its predictive ability. We then recorded for each agent the number of optimal epochs, which is the number of RNN training iterations that minimized underfitting and overfitting. (**a**) We found a strong association between the number of optimal RNN training epochs (y-axis) and the agent's true noise level. This result demonstrates that the number of optimal RNN epochs can be used as a proxy for the amount of information in a certain dataset (given a fixed number of observations and network size). (**b**) For illustration purposes, we plotted the RNN loss curves for agents with low vs. high levels of noise. Specifically, we present loss curves where for each RNN training epoch (x-axis) we estimate the predictive accuracy using a validation dataset. We estimated the loss curves of 300 artificial agents (from three theoretical models) and divided them into two groups according to their true noise (high vs. low). We show that the point of optimal epoch (early stopping; denoted yellow star) was higher to agents with low internal noise.

The online version of this article includes the following figure supplement(s) for figure 3:

**Figure supplement 1.** Correlation matrix between optimal epoch and the true noise.

**Figure supplement 2.** Optimal epoch relation to true noise.

**Figure supplement 3.** Associations between all model parameters and the optimal number of epochs.

behavior of the low IQ participants' part. In that case, RNN should have a similar contribution to the predictive gap, regardless of IQ (see *Figure 4b*).

Furthermore, we estimate the number of optimal epochs for each individual and examine the correlation with IQ. If low IQ participants' behavior is less predicted by the theoretical model mostly due to them being misspecified, then we should not observe any correlation between the optimal number of epochs and IQ (since across IQ participants are assumed to have a similar amount of information in their behavioral data). However, if the behavior of low IQ participants is noisier compared with high IQ, then we should expect a negative correlation between the number of epochs and IQ.

## Predictive performance

We began our investigation by estimating the correlation between IQ and the predictive score of the hybrid model. We found a positive correlation between individuals' IQ and the hybrid predictive score [see *Figure 5a*; linear correlation coefficient $r = 0.28$, p>0.05, 95% CI $(0.014, 0.548)$] so that lower IQ

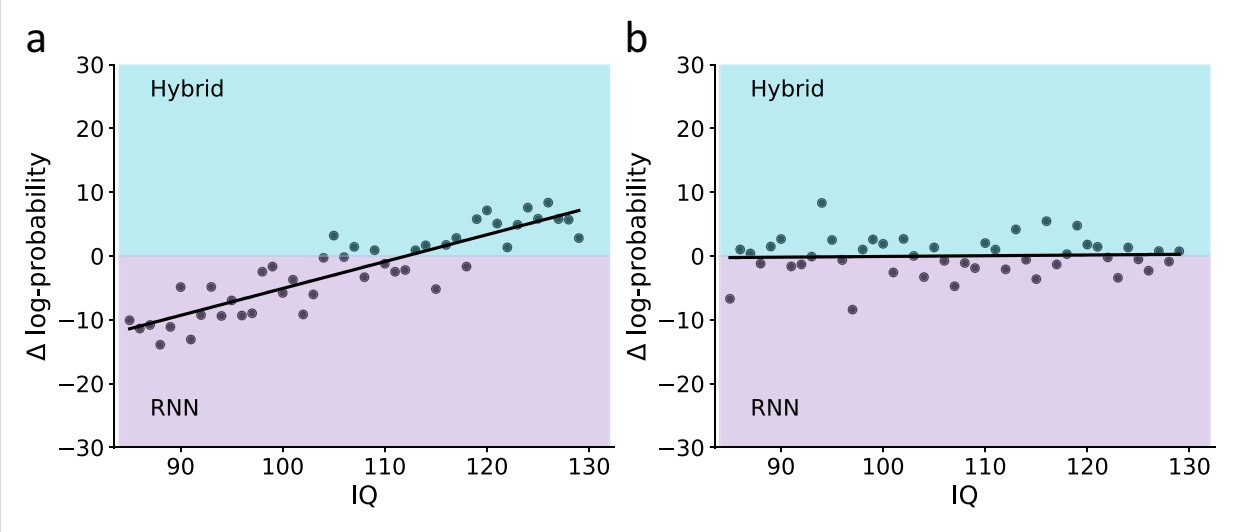

**Figure 4.** Prediction scenarios for the use of recurrent neural network (RNN) to examine model misspecification as a function of IQ. To demonstrate the ability to use RNN to identify model misspecification, we illustrate two hypothetical associations between IQ and the difference in predictive performance for a theoretical hybrid model vs. RNN. (**a**) Prediction for a scenario where there is a higher frequency of model misspecification among low vs. high IQ individuals: here, we assume that low IQ participants are more frequently misspecified by the theoretical hybrid model compared with high IQ individuals. Therefore, the prediction here is that for low IQ individuals the RNN will provide higher predictive gap improvement compared with high IQ individuals. (**b**) Prediction for a scenario where there is equal frequency of model misspecification among low vs. high IQ individuals: here, we assume that the frequency of model misspecification is similar across levels of IQ. Therefore, we predict no association between IQ and the change in predictive accuracy for RNN vs. the theoretical hybrid model.

was associated with a higher predictive gap. This finding can reflect a systematic misspecification of low IQ individuals by the hybrid model. Alternatively, it might be that low IQ is associated with noisy behavior, which then led to a higher predictive gap. To address this question, we predicted individuals' actions using RNN and examined whether the association between individuals' predictive gap and IQ is attenuated when using RNN. If the association between IQ and predictive gap of the theoretical hybrid model is due to higher rates of model misspecification for low compared with high IQ individuals, then RNN would attenuate this association. Specifically, RNN should be able to overcome the misspecification issue the hybrid model might have, thus leading to a similar predictive gap across IQ levels. However, if the association between IQ and predictive gap of the theoretical hybrid model is mostly due to noisy behavior in low compared with high IQ, then we should observe the same correlation between predictive gap and IQ for both the theoretical hybrid model and RNN (see *Figure 5b*, linear correlation coefficient $r = 0.07$, p=0.58).

To examine whether RNN's predictive accuracy shows an attenuated association with IQ compared to the theoretical hybrid predictive accuracy, we estimated the paired interaction of model type (RNN vs. Hybrid) and IQ on predictive accuracy estimates. Specifically, we fitted a hierarchical Bayesian regression model, where the dependent variable was the individual's predictive score (measured in negative log probability) that was predicted using the individual's IQ score, model type (coded as –1 for hybrid and 1 for RNN), and their paired interaction. We found a main effect for IQ, such that higher IQ individuals had higher log probability scores (see *Figure 5—figure supplement 1a*; posterior median = 0.614; 95%$_{HDI}$ between 0.022 and 1.208; probability of direction [pd] 97.9%). Moreover, we also found a main effect for model type, such that the RNN model obtained on average a higher predictive score than the hybrid (see *Figure 5—figure supplement 1b*; posterior median = 9.577; 95%$_{HDI}$ between 6.94 and 12.2; pd 100%).

Importantly, we did not find support for an interaction effect of IQ × model type on individuals' predictive scores, suggesting that the association between IQ and predictive accuracy was similar for the RNN and hybrid models (see *Figure 5c*; posterior median = –0.11; 95% HDI between –0.35 and 0.13; pd 81.1%). This result suggests that RNN did not improve the predictive accuracy of the hybrid model more for low vs. high IQ individuals. Specifically, the lack of interaction between model type and IQ can be taken as evidence that the frequency of model misspecification is not different in low

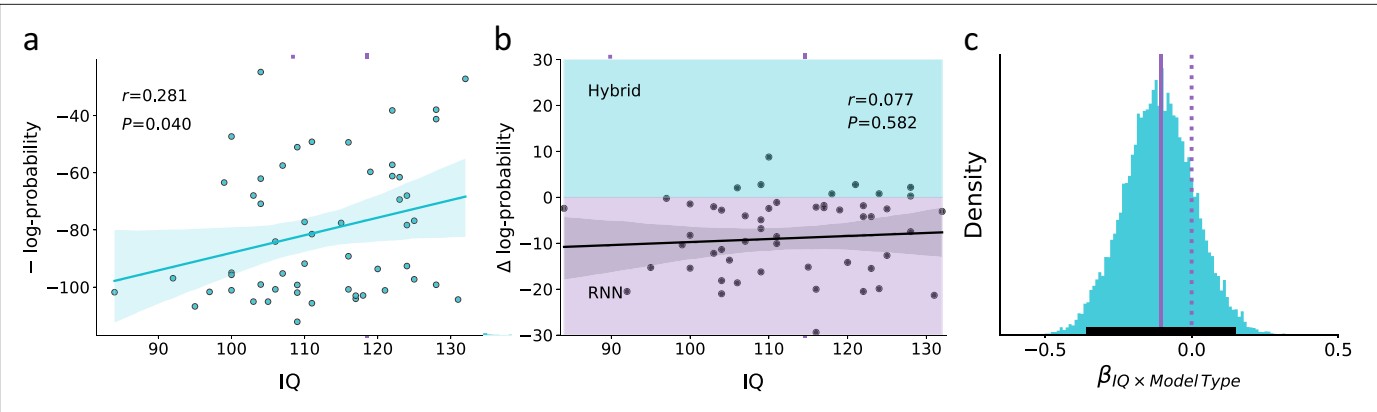

**Figure 5.** Empirical association of the predictive score for the theoretical hybrid model, recurrent neural network (RNN), and IQ. (**a**) Association between IQ and the predictive score obtained from a theoretical hybrid model, showing that low IQ individuals are associated with lower predictive accuracy (shaded area corresponds to the 95% confidence interval for the regression line). We assumed that if this association is mostly due to higher rates of model misspecification in low compared with high IQ individuals, then RNN should overcome this difficulty and show similar predictive accuracy across both models. (**b**) Difference in the predictive score for the theoretical hybrid model vs. RNN. Across different IQ scores, we found no significant difference in the predictive score of one of the models over the other. This finding suggests that the lower predictive score of low IQ individuals is not due to systematic model misspecification, but rather due to noisier behavior (shaded turquoise/purple areas signify better predictive score of the Hybrid/RNN models, respectively; shaded dark area corresponds to the 95% confidence interval for the regression line). (**c**) Posterior distribution for the interaction term from a Bayesian regression, suggesting no effect for the paired interaction of model type (hybrid vs. RNN) and IQ on the model's predictive score. This null effect suggests that the association between IQ and predictive accuracy was similar for both hybrid and RNN models (soiled/dashed purple lines indicate median/zero, respectively; lower solid black line indicates 95%$_{HDI}$). These results suggest that the higher predictive gap of the theoretical model for low compared with high IQ individuals is mostly due to individual differences in the levels of behavioral noise rather than systematic model misspecification.

The online version of this article includes the following figure supplement(s) for figure 5:

**Figure supplement 1.** Posterior distribution for the regression coefficient.

vs. high IQ individuals. To further assert our finding, we now examine the association between IQ and the number of optimal RNN epochs as a proxy for the amount of noise in the individual's data.

## Optimal epoch relation with IQ

The comparison of the predictive performance of the hybrid vs. RNN models suggested that low IQ individuals showed noisier behavior compared to their high IQ peers. To further validate our finding that lower IQ individuals' choice data is noisier (rather than more frequently misspecified), we examined the number of optimal RNN training epochs for each participant. Here, we trained an individual RNN model for each subject initialized with random weights. We then examined the relationship between an individual's IQ score and their corresponding number of optimal RNN training epochs (i.e., the point at which we stopped training the RNN to avoid overfitting). We found a significant positive correlation, such that higher IQ individuals had a higher number of optimal training epochs [see *Figure 6a*; linear correlation coefficient $r = 0.33$, p<0.05, 95% C (0.072, 0.597)]. To further illustrate this finding, we divided participants into two groups according to their IQ score – low/high (using the median as the threshold). We then recorded the prediction of the validation data of each participant throughout the training procedure and averaged within each of the two groups (see *Figure 6b*). We found that the optimal epoch of the high IQ group was significantly greater ($M = 332.80$, $SD = 178.06$) than that of the low IQ group ($M = 227.34$, $SD = 206.90$; $t(52) = 2.011$, p<0.05). These findings suggest noisier behavior for low compared to high IQ individuals.

## Discussion

Developing a theoretical computational model requires the researcher to state theoretical assumptions regarding the investigated process, assumptions that might differ from the true data-generating process, and thus exposed to the problem of model misspecification. Here, we sought to construct a method that tackles this problem in the context of theoretical computational models of human

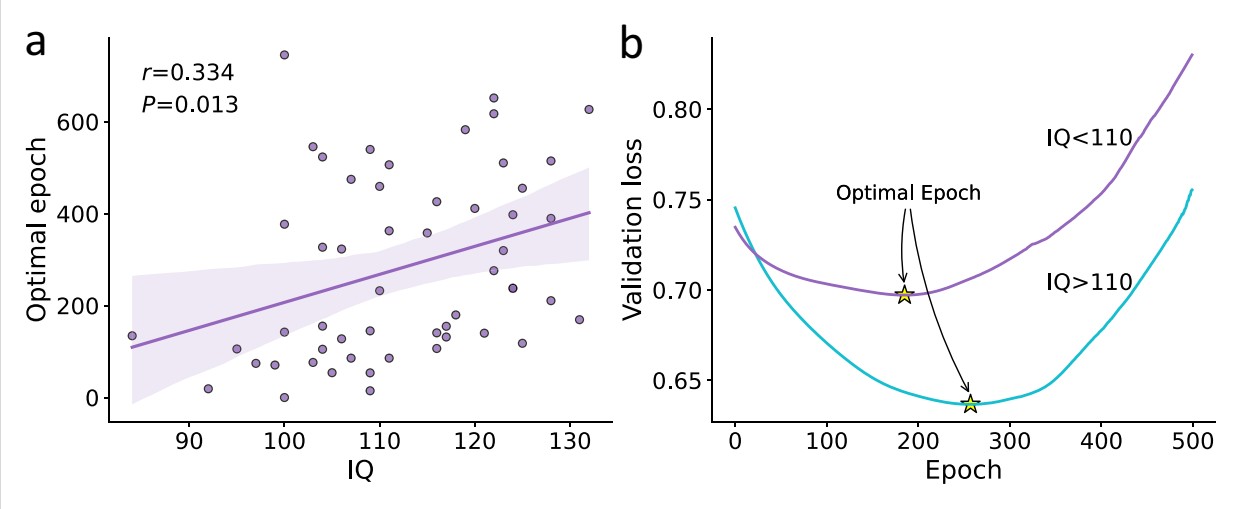

**Figure 6.** Association between IQ and individual's optimal epoch estimates. (**a**) Relation between the number of optimal recurrent neural network (RNN) training epochs (y-axis) and IQ score (x-axis; shaded area corresponds to the 95% confidence interval for the regression line). We found that IQ significantly correlates with individuals' optimal RNN epoch estimates, such that lower IQ participants required fewer RNN training epochs to reach the optimal training point. (**b**) Loss curve of validation data averaged over the low IQ group (IQ < 110; purple) and high IQ group (IQ > 110; turquoise). The point of optimal epoch (early stopping) is denoted by a yellow star. We found that the optimal epoch for the high IQ group was significantly higher than that for the low IQ group. Overall, when combined with our simulation study demonstrating the association between the number of optimal RNN epochs and true noise (for a fixed number of observations and network size; see *Figure 3*), these results suggest noisier decision-making for low IQ individuals compared to high IQ individuals. This finding, along with the results obtained from the RNN predictive accuracy (see *Figure 5*), suggests that low IQ individuals are not more frequently misspecified compared to their high IQ peers.

decision-making behavior. Taking advantage of the high flexibility of theory-independent models (i.e., RNN) that are not theoretically constrained by assumptions of behavior, we proposed a method to indicate if a predictive gap observed in a single-agent choice data is mostly due to model misspecification or rather an inherent stochasticity in the behavior being modeled. We further suggest that the number of training epochs used to reach an optimal balance between under and overfitting for RNN can be used as a relative estimate of noise in the individual behavior.

In Study 1, we validated our approach using simulated data from artificial agents generated by different theoretical models in a two-step multi-armed bandit task. By comparing the predictive performance of the RNN model with each theoretical model, we demonstrated the RNN's ability to determine whether a predictive gap observed in an individual agent is primarily due to model misspecification or inherent stochasticity in their behavior. Additionally, we showed that the point of early stopping in the RNN training process is strongly associated with the level of true irreducible stochasticity in an agent's behavior. A higher number of optimal training epochs is indicative of less noise in the agent's behavior, given a fixed amount of data and network size. One practical advantage of using the point of early stopping is that it can be estimated using only two sets of data (training and validation), unlike predictive accuracy, which requires an additional test dataset.

Next, in Study 2, we applied these methods to an empirical dataset of human participants performing the same two-step task. We found that Daw's hybrid model (*Daw et al., 2011*), a well-known theoretical model, exhibited a consistently poorer fit to participants with low IQ compared to those with high IQ. To investigate the source of this poorer fit, we individually fitted an RNN model to each participant and compared the predictive performance of both the hybrid model and the RNN model, as well as the point of early stopping for each participant. We observed that the RNN model showed a similar reduction in behavior prediction for both low and high IQ individuals, consistent with the hybrid model. Additionally, the RNN's optimal epoch estimates were systematically higher for high IQ individuals. These findings provide preliminary evidence suggesting that the behavior of low IQ participants is characterized by greater inherent noise, rather than being more misspecified by the hybrid model, when compared to their high IQ counterparts.

The use of theory-independent neural network models fitted directly to behavioral data in decision-making tasks has garnered significant attention in recent years (*Dezfouli et al., 2019a*; *Dezfouli*

*et al., 2019b*; *Fintz et al., 2022*; *Peterson et al., 2021*; *Song et al., 2021*). Previous research has highlighted the advantages of this approach by demonstrating that RNNs are capable of capturing behavioral features that traditional theoretical models struggle to replicate. In a groundbreaking study by *Dezfouli et al., 2019b*, it was shown that an RNN model fitted to participants performing a two-armed bandit task outperformed a typical RL model in pure action prediction. This finding has been extended to more complex decision-making tasks such as the four-armed bandit (*Fintz et al., 2022*) and build-your-own-stimulus tasks (*Song et al., 2021*). While most studies have primarily focused on the high predictive performance of RNN models, there are also examples of leveraging RNNs to gain theoretical insights. For instance, *Dezfouli et al., 2019a* developed an autoencoder model utilizing an RNN encoder to map each participant's behavior into a low-dimensional latent space, which facilitated the study of individual differences and variation in decision-making strategies. In another study, *Song et al., 2021* proposed an RNN model trained with an embedding specific to each participant, showing that these embeddings were effective in capturing differences in various cognitive variables. Thus, previous research has demonstrated that RNNs, unconstrained by specific theoretical assumptions, have the ability to capture and predict human behavior, resulting in remarkably low predictive gaps.

We believe that our work contributes to the growing body of research in the field, making four key contributions. First, through simulation, we have validated the assertion that RNNs can achieve nearly optimal fit and approximate various behavioral models without the need for prior theoretical specification. Second, we introduce a novel application of the point of early stopping in RNN training. Traditionally used to prevent overfitting, we demonstrate its utility as a unique individual measure of the amount of inherent noise present in both artificial and human behavioral choice data. Unlike predictive accuracy, which requires group-level pre-training and three datasets per individual (training, validation, and test), the point of early stopping only necessitates two datasets (training and validation) and can be employed without empirical RNN pre-training. This allows for a more flexible estimate of stopping points between individuals. Third, we present an application of the RNN model to human choice data in the context of a multistage two-step decision task, expanding upon previous research that has predominantly focused on single-stage tasks. Lastly, we utilize the RNN model to investigate the relationship between model misspecification and individuals' intelligence scores.

Our work also addresses the question of inherent randomness or irreducible stochasticity in human behavior. The question of irreducible stochasticity has broader implications not only for cognitive science but also as a fundamental question in general science. We provide provisional evidence suggesting that individuals with lower intelligence scores exhibit noisier decision patterns that cannot be reduced by any model. However, several questions remain unanswered, such as identifying the specific stages of the decision-making process that underlie this behavioral variability. In the current study, we primarily focus on noise that arises from the deliberation process. This aligns with previous studies that have proposed the concept of 'decision acuity' (*Moutoussis et al., 2021*), which represents a dimensional structure in core decision-making strongly related to the inverse temperature parameter ($\beta$). It suggests that decision variability originates from differences in reward sensitivities. However, another line of research has proposed the idea of 'computation noise', which refers to random noise inherent to the learning process that corrupts the value updating of each action (*Findling et al., 2019*; *Findling and Wyart, 2021*). Further studies are needed to elucidate the extent to which these two factors contribute to the noisier decision patterns observed in individuals with lower intelligence. Another open question pertains to identifying the neural correlates and mechanisms underlying this variability. Nevertheless, the significant advantage of the current method is that researchers can use the number of optimal RNN epochs to estimate the amount of noise in observations without specifying a theoretical mechanism. This capability enhances the interpretation and utilization of theoretical models.

Several limitations should be considered in our proposed approach. First, fitting a data-driven neural network is evidently not enough to produce a comprehensive theoretical description of the data generation mechanisms. Currently, best practices for cognitive modeling (*Wilson and Collins, 2019*) require identifying under what conditions the model struggles to predict the data (e.g., using posterior predictive checks), and describing a different theoretical model that could account for these disadvantages in prediction. However, identifying conditions where the model shortcomings in predictive accuracy are due to model misspecifications rather than noisier behavior is a challenging task. We

propose leveraging data-driven RNNs as a supplementary tool, particularly when they significantly outperform existing theoretical models, followed by refined theoretical modeling to provide insights into what processes were misspecified in the initial modeling effort.

Second, although we observed a robust association between the optimal number of epochs and true noise across varying network sizes and dataset sizes (see *Figure 3—figure supplement 1*), additional factors such as network architecture and other model parameters (e.g., learning rate, see *Figure 3—figure supplement 3*) might influence this estimation. Further research is required to allow us to better understand how and why different factors change the number of optimal epochs for a given dataset before it can be applied with confidence to empirical investigations.

Third, the empirical dataset used in our study consisted of data collected from human participants at a single time point, serving as the training set for our RNN. The test set data, collected with a time interval of ~6 and 18 months, introduced the possibility of changes in participants' decision-making strategies over time. In our analysis, we neglected any possible changes in participants' decision-making strategies during that time, changes that may lead to poorer generalization performance of our approach. Thus, further studies are needed to eliminate such possible explanations.

Fourth, our simulations, albeit illustrative, were confined to known models, necessitating in silico validation before extrapolating the efficacy of our approach to other model classes and tasks. Our aim was to showcase the potential benefits of using a data-driven approach, particularly when faced with unknown models. However, whether RNNs will provide optimal fits for tasks with more complex rules and long-term sequential dependencies remains uncertain.

Finally, while positive outcomes where RNNs surpass theoretical models can prompt insightful model refinement, caution is warranted in directly equating RNN performance with that of the generative model, as seen in our simulations (e.g., *Figure 2*). We highlight that our empirical findings depict a more complex scenario, wherein the RNN enhanced the predictive accuracy for all participants uniformly. Notably, we also provide evidence supporting a null effect among individuals, with no consistent difference in RNN improvement over the theoretical model based on IQ. Although it remains conceivable that a different data-driven model could systematically heighten the predictive

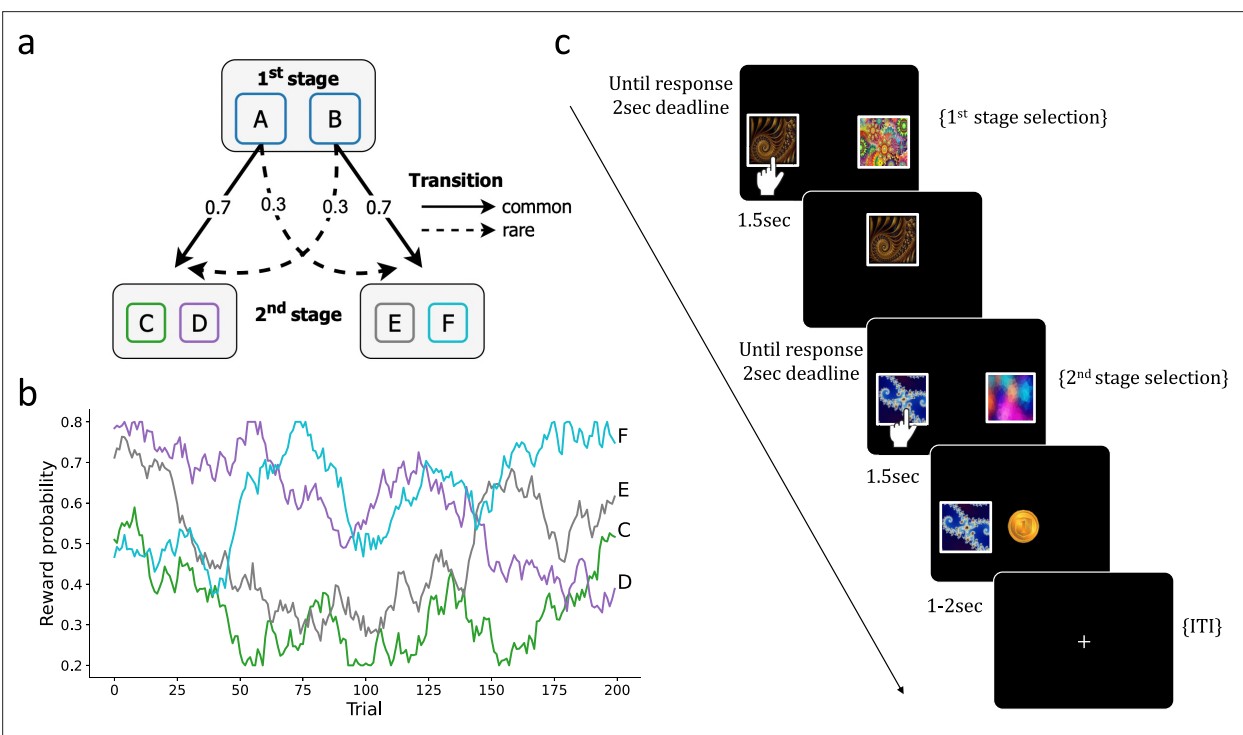

**Figure 7.** Two-step task. (**a**) At the first stage, participants choose between two options (A or B) that probabilistically lead to one of two second-stage states, with a fixed transition probability of 70% ('common') or 30% ('rare'). In the second stage, participants choose between two options (C/D or E/F) to obtain rewards. (**b**) The reward probability for each second-stage choice is determined by a random walk drift. (**c**) An example of a trial sequence in the two-step task.

accuracy for lower IQ individuals in this task, such a possibility seems less probable in light of the current findings.

To sum up, we show that both the predictive gap and the point of early stopping of neural networks can be used to estimate whether a certain predictive gap found for a theoretical model is mostly due to model misspecification or irreducible noise. We hope this work will lead to further studies exploring the utilization of neural networks to enhance theoretical computational models.

## Materials and methods
### Study 1: Simulation study
#### Two-step task
To test our hypothesis, we employed an exemplar two-step RL task, which has been widely used in computational modeling studies (see *Figure 7a*; *Daw et al., 2011*). Each trial of the task consisted of two stages. In the first stage, participants made a choice between two actions, labeled as action A and action B. These actions probabilistically led to one of two possible second-stage states. Action A predominantly led to State I in the second stage, while action B primarily led to State II (common transition). However, there was also a minority of trials where the actions led to the opposite states, meaning action A led to State II and action B led to State I (rare transition). The probabilities of common and rare transitions were set at 0.7 and 0.3, respectively, and remained constant throughout the task. In the second stage, participants again made a choice between two additional actions and received a binary reward of $0 (no reward) or $1 (reward). The reward probability varied randomly across trials (see *Figure 7b*).

#### Theoretical models
We consider three different theoretical models that generate choice behavior in the two-step task.

#### Hybrid model
The first model we considered was the hybrid model originally suggested by *Daw et al., 2011*, which assumes that agents' choice behavior is determined by a weighted combination of both model-based (MB) and model-free (MF) RL algorithms (*Sutton and Barto, 2018*). The contribution of each algorithm is modulated by the free parameter weight $w$. The weighted value of each first-stage action is calculated according to

$$Q^{net}(s_1, a_1) = w \cdot Q^{MB}(s_1, a_1) + (1 - w) \cdot Q^{MF}(s_1, a_1), \tag{1}$$

where $Q^{MF}(s_1, a_1)$ is the MF action value in the first stage updated trial-by-trial and $Q^{MB}(s_1, a_1)$ is the mentally calculated MB estimation. The MF Q-values were initiated at zero and updated according to action and reward history as follows:

$$Q^{MF}(s_1, a_1) = Q^{MF}(s_1, a_1) + \alpha_1 \cdot \left( Q^{MF}(s_2, a_2) - Q^{MF}(s_1, a_1) \right) + \alpha_1 \cdot \lambda \cdot \left( r_t - Q^{MF}(s_2, a_2) \right), \tag{2}$$

$$Q^{MF}(s_2, a_2) = Q^{MF}(s_2, a_2) + \alpha_2 \cdot \left( r_t - Q^{MF}(s_2, a_2) \right), \tag{3}$$

where $\alpha_1$, $\alpha_2$ are the learning rates free parameters of the first and second stages, respectively, $\lambda$ is the discount factor parameter, and $r_t \in [0, 1]$ is the received reward. The MB action value of the first stage $Q^{MB}(s_1, a_1)$ is calculated according to

$$Q^{MB}(s_1, a_1) = \sum_{s_2 \in S} P(s_2 | s_1, a_1) \max_{a_2 \in A} Q^{MB}(s_2, a_2), \tag{4}$$

where $P(s_2 | s_1, a_1)$ is the true transition probability of reaching the second-stage state $s_2$ by performing action $a_1$ at the first stage. $S$ denotes the second-stage states and $A$ is the set containing the actions available at each second-stage state, respectively. Note that at the second stage MB values, $Q^{MB}(s_2, a_2)$ of each action $a_2$ performed at the second-stage state $s_2$ are identical to the corresponding MF values, namely, $Q^{MB}(s_2, a_2) = Q^{MF}(s_2, a_2)$. We used a SoftMax choice rule for both the first and second choices. The SoftMax choice rule transforms the action values to distribution over the action according to

$$P(s_1, a_1) = \frac{\exp\left[\beta_1 \cdot Q^{net}\left(s_1, a_1\right)\right]}{\sum_{\tilde{a} \in A} \exp\left[\beta_1 \cdot Q^{net}\left(s_1, \tilde{a}\right)\right]}, \tag{5}$$

$$P(s_2, a_2) = \frac{\exp\left[\beta_2 \cdot Q^{MF}\left(s_2, a_2\right)\right]}{\sum_{\tilde{a} \in A} \exp\left[\beta_2 \cdot Q^{MF}\left(s_2, \tilde{a}\right)\right]}, \tag{6}$$

where $\beta_1$ and $\beta_2$ are inverse-temperature parameters of the first and second stages, respectively. This parameter controls the level of stochasticity in the action selection, where $\beta \to 0$ corresponds to random choices and $\beta \to \infty$ for deterministically choosing the highest value action. Overall, the model includes six free parameters $\theta_{hybrid} = \{w, \alpha_1, \alpha_2, \lambda, \beta_1, \beta_2\}$.

## Habit model

According to a habit model, agents tend to repeat previously taken actions regardless of their outcome (*Miller et al., 2019*). Here, agents' choices are influenced only by previous actions, independent of any transition or reward history. The model keeps track of past action selection in the form of habit strengths, denoted as $H$, and are similar in spirit to the $Q$-values described in the previous hybrid model. These values are initiated at 0.5 and updated on both stages of the task according to

$$\mathrm{H}(\mathrm{s}_1, a) = \begin{cases} H(s_1, a_i) + \alpha_1 \cdot \left(1 - H(s_1, a_i)\right) \\ H(s_1, a_j) + \alpha_1 \cdot \left(0 - H(s_1, a_j)\right) \end{cases}, \tag{7}$$

$$\mathrm{H}(\mathrm{s}_2, a) = \begin{cases} H(s_2, a_i) + \alpha_2 \cdot \left(1 - H(s_2, a_i)\right) \\ H(s_2, a_j) + \alpha_2 \cdot \left(0 - H(s_2, a_j)\right) \end{cases}, \tag{8}$$

where $\alpha_1$, and $\alpha_2$ are the learning rates free parameters of the first and second stages, respectively. $H(s, a)$ is the habit strengths matrix of all possible states, and $a$ is the vector of all actions available in the corresponding state. We denote the action selected in the current trial as $a_i$ and the unchosen action as $a_j$. The updating rule increases the value of the action selected ($a_i$) toward 1 and the other unselected action ($a_j$) toward 0. Like the hybrid model, we used SoftMax choice rule for both the first and second choices:

$$P(s_1, a_1) = \frac{\exp\left[\beta_1 \cdot H\left(s_1, a_1\right)\right]}{\sum_{\tilde{a} \in A} \exp\left[\beta_1 \cdot H\left(s_1, \tilde{a}\right)\right]}, \tag{9}$$

$$P(s_2, a_2) = \frac{\exp\left[\beta_2 \cdot H\left(s_2, a_2\right)\right]}{\sum_{\tilde{a} \in A} \exp\left[\beta_2 \cdot H\left(s_2, \tilde{a}\right)\right]}, \tag{10}$$

Overall, the model includes four free parameters $\theta_{habit} = \{\alpha_1, \alpha_2, \beta_1, \beta_2\}$.

## K dominated-hand model (K-DH)

In this model, an action is selected in a random fashion with some bias toward one action over the other (e.g., due to hand dominancy; note that in the simulation we assigned each action to the same response key, thus a bias toward one response key translates directly to a tendency to choose one action over the alternative). In this model, the agent is assumed to change the preference for the dominant hand across a sequence of $K$ actions. For example, in $K = 2$ the agent will have two probabilities (defined by two fixed parameters; $p_1$, $p_2$) representing the preference for the dominant hand as a function of position in the two trials' sequence. The K-dominant hand model is therefore designed to generate a random sequence of choices (e.g., A, B, A, B …) with some systematic repetitions. Here, we included agents with a fixed $K = 2$ for the first stage and $K = 1$ for each of the two-second stages, which resulted in a similar number of free parameters as other models in this work. Specifically, the probability of selecting action $a_i$ at each state of the task is given by

$$p(s_1, a_i) = p_{t \bmod 2}, \tag{11}$$

$$p(s_2, a_i) = p_2, \quad p(s_3, a_i) = p_3, \tag{12}$$

where $p_0/p_1$ are the probabilities of selecting action $a_i$ in the first stage on the even/odd trials, respectively, and $p_2/p_3$ are the probabilities of selecting action $a_i$ at each second stage state, respectively. The probability of selecting the other action at each state is always set to be the complementary probability (i.e., $1 - p(s, a_i)$). Overall, the model includes four free parameters $\theta_{K-DH} = \{p_0, p_1, p_2, p_3\}$.

## Theory-independent models

We used two theory-independent models. Unlike the theoretical models which are used both to generate, fit, and predict choice data, the theory-independent models were used only to fit and predict choice data.

### Recurrent neural network (RNN)

The first theory-independent model we considered was a three-layer RNN architecture. Our RNN consisted of an (i) input layer, (ii) hidden layer, and (iii) output layer. At each step, the RNN input was the last trial's first-stage action, reward, and transition type. The output at each step is a probability distribution over the current trial's first-stage actions. The hidden layer was based on a single-layer gated recurrent unit (GRU; *Cho et al., 2014*) with five hidden units.

$$\mathbf{x}_t = [r_{t-1}, a_{t-1}, t_{t-1}], \tag{13}$$

$$\mathbf{r}_t = \sigma(\mathbf{W}_r \mathbf{x}_t + \mathbf{U}_r \mathbf{h}_{t-1} + \mathbf{b}_r), \tag{14}$$

$$\mathbf{z}_t = \sigma(\mathbf{W}_z \mathbf{x}_t + \mathbf{U}_z \mathbf{h}_{t-1} + \mathbf{b}_z), \tag{15}$$

$$\hat{h}_t = \phi(W_h x_t + U_h(r_t \odot h_{t-1}) + b_h), \tag{16}$$

$$\mathbf{h}_t = (1 - \mathbf{z}_t) \odot \mathbf{h}_{t-1} + \mathbf{z}_t \odot \hat{\mathbf{h}}_t, \tag{17}$$

$$\mathbf{o}_t = \mathbf{W}_o \mathbf{h}_t + \mathbf{b}_o, \tag{18}$$

$$p(a_t = A) = \frac{e^{\mathbf{o}_t}}{\sum_i e^{\mathbf{o}_i}}, \tag{19}$$

where $\mathbf{x}_t$ is the input vector and $\mathbf{h}_{t-1}$ is the previous hidden state vector. $\sigma$ is the logistic sigmoid function, $\odot$ is the Elementwise Hadamard product, $\phi$ is the hyperbolic tangent function. $r_t$ represents the reset gate, $z_t$ the update gate, and $h_t$ the candidate activation vectors. $\mathbf{W}_r, \mathbf{W}_z, \mathbf{W}_h$ are the input connection weight matrices, $\mathbf{U}_r, \mathbf{U}_z, \mathbf{U}_h$ are the recurrent connection weight matrices, and $\mathbf{b}_r, \mathbf{b}_z, \mathbf{b}_h$ are the bias vectors. $h_t$ represents the actual activation vector of the unit (current hidden state). $h_t$ is projected to the output layer with full connections via the weight matrix $\mathbf{W}_o$ and bias vector $\mathbf{b}_o$. The output is then transformed with a SoftMax activation function to action probabilities. Overall, this model includes the following free parameters: $\theta_{\mathrm{RNN}} = \{\mathbf{W}_r, \mathbf{W}_z, \mathbf{W}_o, \mathbf{U}_r, \mathbf{U}_z, \mathbf{U}_h, \mathbf{b}_r, \mathbf{b}_z, \mathbf{b}_h, \mathbf{W}_o, \mathbf{b}_o\}$. The total number of free parameters is dependent on the size of the hidden layer (192 for five hidden units). Importantly, in the supplementary information, we provide additional analysis where we varied the number of hidden units of the hidden GRU layer (2 and 10). We found similar results, suggesting that at least with the current task, our approach is not sensitive to the size of the network (see *Figure 2—figure supplement 2*)

### Logistic regression (LR)

The second theory-independent model we considered was an LR model, where the probability of taking the first-stage action is determined by a linear combination of the history of past actions, rewards, and transition, up to $j$ trials back.

$$x = \beta_0 + \sum_j \left( \beta_a^j a_{t-j} + \beta_r^j r_{t-j} + \beta_t^j t_{t-j} + \beta_{a \times r}^j a_{t-j} r_{t-j} + \beta_{a \times t}^j a_{t-j} t_{t-j} + \beta_{r \times t}^j r_{t-j} t_{t-j} + \beta_{a \times r \times t}^j a_{t-j} r_{t-j} t_{t-j} \right),$$

$$\tag{20}$$

$$p(a_t = 1 \mid \beta) = \frac{1}{1 + e^{-x}}, \tag{21}$$

The dependent and independent variables were coded as follows: $a_t$: 1 for action A and –1 for action B (first-stage actions, see *Figure 7*). $a_{t-j}$: 0.5 if the first state action $j$ trials back was A and –0.5 otherwise. $r_{t-j}$: 0.5 if $j$ trials back was rewarded and –0.5 otherwise. $t_{t-j}$: 0.5 if $j$ trials back transition was common and –0.5 if it was rare. Overall, the model includes $(1 + j \times 7)$ free parameters $\theta_{\mathrm{LR}} = \{\beta_0, \beta_a^j, \beta_r^j, \beta_t^j, \beta_{a \times r}^j, \beta_{a \times t}^j, \beta_{r \times t}^j, \beta_{a \times r \times t}^j\}$.

**Table 1.** Simulation specification for Study 1.

| Model | Hybrid | Habit | K-DH |
|---|---|---|---|
| Parameter range | $w \sim U(0,1)$ | | |
| | $\alpha_{1,2} \sim U(0,1)$ | $\alpha_{1,2} \sim U(0,1)$ | $p_{0,1,2,3} \sim U(0,1)$ |
| | $\beta_{1,2} \sim U(0,10)$ | $\beta_{1,2} \sim U(0,10)$ | |
| | $\lambda \sim U(0,1)$ | | |
| # agents | 100 | 100 | 100 |
| # trials | 200 | 200 | 200 |
| # blocks | 3 | 3 | 3 |

## Simulating data and model fitting

First, we simulated choice behavior of artificial agents on the two-step task (see *Figure 7*) using the three distinct theoretical data-generating models (Hybrid, Habit, and K-DH). For each model, we simulated 100 agents with their true underlying free parameters sampled from a range of possible options (see *Table 1*). Each agent was simulated for three blocks of 200 trials each. Then, we separately fitted all five models (three theoretical and two theory-independent) to data simulated from all three theoretical models. In the supplementary information, we provide an additional analysis where we varied the lengths of each block (100 and 500 trials) and found similar results suggestive that our method (at least for the current task) is not sensitive to the amount of data (see 'Code availability').

### Theoretical models

For each agent individually, we sought optimal parameters $\hat{\theta}_m$ under each of the three theoretical models (Hybrid, Habit, and K-DH) with maximum-likelihood estimation using only one of the agents' blocks (training-set). We repeated this procedure with 10 different initial search points and chose the optimal $\hat{\theta}_m$ using a withheld block (validation set). Finally, the optimal parameters $\hat{\theta}_m$ were used to record the prediction on an additional withheld block (test set). We used SciPy library (*Virtanen et al., 2020*) with the minimize function and L-BFGS-B method to extract best-fit parameters.

### Recurrent neural network

We trained an individual RNN for each agent in a supervised manner with cross-entropy loss (maximum-likelihood) using Pytorch library (*Paszke et al., 2019*). We trained with Adam optimizer (*Kingma and Ba, 2014*) with a constant learning rate of 0.001. Crucially, unlike the other models, RNN's high flexibility may lead it to overfit the training data. This overfitting of the training dataset will result in a low generalization, making the model less useful for making predictions on new data. To prevent this, we used early stopping, a commonly used regularization method (*Bishop and Nasrabadi, 2006*). We therefore first pre-trained the RNN model using a new auxiliary synthetic dataset consisting of 200 agents from all three theoretical models pooled together. For the auxiliary synthetic data, agents were simulated for two blocks of 200 trials each (used as training and validation for generating the RNN pre-training). The pre-trained RNN model was then fine-tuned for each individual on the main synthetic dataset (see Simulating data and model fitting) using early stopping. Specifically, we used the three blocks of each agent as a train, validation (used for training using early stopping), and a test set used to estimate the predictive accuracy that we report in the results section for further analysis. To further allow an accurate and variable possible estimate of early stopping, we recorded the point of early stopping (using training and validation sets) for an individual RNN for each agent that was not pre-trained, and instead initialized with random weights. The point at which we stopped optimizing the network (i.e., the number of training epochs) was then used in the main analysis and referred as *optimal epoch* estimate.

### Logistic regression

We fitted an LR model to the artificial data using scikit-learn library (*Pedregosa et al., 2011*) with L-BFGS-B method, again using only one of the agents' blocks (training set). The individual coefficient

of each agent was estimated using maximum-likelihood. The number of trials back $k$ was determined using a withheld block (validation set). Then we used the coefficient to predict the choices of a withheld block (test et).

## Model selection

We compare the different models by their predictive performance of left-out data of the first-stage choice. Hence, we adopted a CV approach. Specifically, at each CV round, for each agent, and under each model, only one of the blocks (training set) was used to estimate optimal parameters ($\hat{\theta}_m$). Then the optimal parameters were used to evaluate the predictions on a withheld block (test set). We averaged across all withheld blocks (three in total) to obtain a single predictive score we denote as $nLP_i^m$ (negative log probability; lower is better),

$$nLP_i^m = \frac{-\sum_{b=1}^{B} \sum_{t=1}^{T} \log p(a_t^i | \hat{\theta}_m)}{|B|}, \tag{22}$$

where $m$ denotes the fitted model, $i$ the agent index, $B$ the total number of blocks, $T$ the total number of trials of the corresponding block, $\hat{\theta}_m$ denotes the optimal parameters under model $m$ (maximum-likelihood parameters of the validation set), and $\log p(a_t^i | \hat{\theta}_m)$ is the log probability that the optimal parameters under model $m$ assign to the first-stage action $a_t$ of the test set.

## Noise estimates

For each agent, we quantified the amount of true noise (stochasticity) the agent holds in the first-stage action using his true underlying parameters. Specifically, within each group of agents that shared the same theoretical model we performed a min-max scaling. For the hybrid agents, we used the agent's true inverse-temperature parameter of the first-stage $\beta_1$ to calculate the amount of true noise as follows:

$$noise = 1 - \frac{\beta_1^i - \beta^{min}}{\beta^{max} - \beta^{min}}, \tag{23}$$

where $\beta_1^i$ is the true first-stage inverse-temperature of agent $i$ and $\beta^{min}/\beta^{max}$ is the true minimal/ maximal inverse-temperature overall hybrid agents. We took the $1 -$ min-max scaling so that the hybrid agent with the minimal amount of noise will take a value of 0 and the agent with the maximal amount of noise will take the value of 1. For the habit agents, we performed the same calculation using the true inverse-temperature parameter $\beta_1$ of the first stage.

For the K-DH agent, we used the true $p_0$ and $p_1$ parameters to quantify the amount of true noise. We measured the distance (in absolute value) between the true $p_0/p_1$ parameters to 0.5 (i.e., completely random response policy) and summed these resulting distances:

$$\delta = |p_0 - 0.5| + |p_1 - 0.5|, \tag{24}$$

and then performed a min-max scaling as follows:

$$noise = 1 - \frac{\delta^i - \delta^{min}}{\delta^{max} - \delta^{min}}, \tag{25}$$

That is, a K-DH agent that his true $p_0, p_1 = 0$ (i.e., determinedly choose one of the actions at each trial) will take the value 0. Conversely, a K-DH agent that his true $p_0, p_1 = 0.5$ (i.e., choose randomly at each trial) will take the value 1.

## Study 2: Empirical study

### Dataset

We studied a previously published dataset taken from NSPN U-Change Cognition Cohort (*Kiddle et al., 2018*). We focus on a subset of the dataset, of which healthy volunteers (ages 14–24 years; $N = 54$) performed the two-step task at three distinct time points (~6 and 18 months after the first measurement). At the first time point (Time I), each subject performed the task for 121 trials, at the second (Time II) 121 trials, and at the third (Time III) 201 trials. In the preprocessing step, we omitted from the analysis trials with RTs below 150 ms. The dataset also includes for each subject a Wechsler

Abbreviated Scale of Intelligence (*Wechsler, 1999*). Importantly, we also utilized another subset of the NSPN dataset that included $N = 515$ individuals who performed the two-step task in only two different time points (Time I: 121 trials; Time III: 201 trials).

## Model fitting

For each subject choice data, we fitted both the theoretical hybrid model and a theory-independent RNN model. We followed the same procedure presented in Study 1, where at each CV round (three in total), we used each subject three distinct measurements as a train, validation, and test sets, respectively.

## Hybrid model

To comply with most studies that examined two-step task behavior using a hybrid model (*Daw et al., 2011*; *Gillan et al., 2016*; *Shahar et al., 2019*), we included in the hybrid model described in Study 1 an additional first-stage choice perseveration parameter:

$$P(s_1, a_1) = \frac{\exp\left[\beta_1 \cdot \left(Q^{net}(s_1, a_1) + \kappa_t\left(a\right)\right)\right]}{\sum_{\tilde{a} \in A} \exp\left[\beta_1 \cdot \left(Q^{net}(s_1, \tilde{a}) + \kappa_t\left(\tilde{a}\right)\right)\right]}, \qquad \kappa_t\left(a\right) = \begin{cases} \kappa & \text{if } a = a_{t-1} \\ 0 & \text{otherwise} \end{cases}, \qquad (26)$$

where $-0.5 < \kappa < 0.5$ is the choice perseveration free parameter so that $\kappa > 0$ corresponded to selecting the same action as the previous trial and $\kappa < 0$ corresponded to switching to the other action. All other model details were similar to the model presented in Study 1. Overall, the model includes seven free parameters $\theta_{\text{hybrid}} = \{w, \alpha_1, \alpha_2, \lambda, \beta_1, \beta_2, \kappa\}$.

## Recurrent neural network

The RNN architecture was identical to the one described in Study 1 (see 'Theory-independent models') We first pre-trained the RNN using a subset of the NSPN dataset consisting of $N = 515$ individuals who performed the two-step task at two different time points (Time I: 121 trials; Time III: 201 trials). As in Study 1, we used one block (from each individual) for training and the other block for validation. We then fine-tuned the pre-trained RNN weights to fit the choice behavior of each of the $N = 54$ individuals using three different blocks for each individual, following the same CV procedure of Study 1. In this procedure, each subject's three distinct measurements were used as training, validation, and test sets, respectively, in three rounds. We were concerned that it might be that the pre-training set ($N = 515$, two sessions) had a different proportion of high/low IQ participants compared to the main set that we were interested in estimating ($N = 54$, three sessions). However, we tested and found that the distribution of IQ scores was similar for both subsets ($M = 112.31$, $SD = 10.75$ for $N = 54$ and $M = 110.83$, $SD = 10.92$ for $N = 515$; $t\left(567\right) = -0.954$, $p = 0.34$).

## Acknowledgements

The study was funded by Israeli Science Foundation (grant 2536/20 awarded to NS).

# Additional information

## Funding

| Funder | Grant reference number | Author |
|---|---|---|
| Israel Science Foundation | 2536/20 | Nitzan Shahar |

The funders had no role in study design, data collection and interpretation, or the decision to submit the work for publication.

## Author contributions

Yoav Ger, Conceptualization, Data curation, Software, Formal analysis, Investigation, Visualization, Methodology, Writing – original draft, Writing – review and editing; Moni Shahar, Conceptualization,

Formal analysis, Supervision, Investigation, Methodology, Writing – original draft, Writing – review and editing; Nitzan Shahar, Conceptualization, Formal analysis, Supervision, Funding acquisition, Investigation, Methodology, Writing – original draft, Writing – review and editing

### Author ORCIDs
Yoav Ger ⓘ https://orcid.org/0000-0002-4847-0146
Nitzan Shahar ⓘ https://orcid.org/0000-0002-1364-6738

Reviewer #1 (Public review): https://doi.org/10.7554/eLife.90082.3.sa1
Reviewer #2 (Public review): https://doi.org/10.7554/eLife.90082.3.sa2
Author response https://doi.org/10.7554/eLife.90082.3.sa3

## Additional files

### Supplementary files
MDAR checklist

### Data availability
The current manuscript is a computational study, so no data have been generated for this manuscript. The code is publicly available via: https://github.com/yoavger/using_rnn_to_estimate_irreducible_stochasticity (copy archived at *Yoavger, 2023*).

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
