## [Editor Report · eLife assessment]

In this study, Ger and colleagues present a **valuable** new technique that uses recurrent neural networks to distinguish between model misspecification and behavioral stochasticity when interpreting cognitive-behavioral model fits. Simulations provide **solid** evidence for the validity of this technique and broadly support the claims of the article, although more work is needed to understand its applicability to real behavioral experiments. This technique addresses a long-standing problem that is likely to be of interest to researchers pushing the limits of cognitive computational modeling.

---

## [Referee Report · Reviewer #1 (Public review)]

Summary:

Ger and colleagues address an issue that often impedes computational modeling: the inherent ambiguity between stochasticity in behavior and structural mismatch between the assumed and true model. They propose a solution to use RNNs to estimate the ceiling on explainable variation within a behavioral dataset. With this information in hand, it is possible to determine the extent to which "worse fits" result from behavioral stochasticity versus failures of the cognitive model to capture nuances in behavior (model misspecification). The authors demonstrate the efficacy of the approach in a synthetic toy problem and then use the method to show that poorer model fits to 2-step data in participants with low IQ are actually due to an increase in inherent stochasticity, rather than systemic mismatch between model and behavior.

Strengths:

Overall I found the ideas conveyed in the paper interesting and the paper to be extremely clear. The method itself is clever and intuitive and I believe it could potentially be useful in certain circumstances, particularly ones where the sources of structure in behavioral data are unknown. Support for the method from synthetic data is clear and compelling. The flexibility of the method means that it could potentially be applied to different types of behavioral data - without any hypotheses about the exact behavioral features that might be present in a given task.

Weaknesses:

That said, I have some concerns with the manuscript in its current form, largely related to the applicability of the proposed methods for problems of importance in computational cognitive neuroscience. This concern stems from the fact that the toy problem explored in the manuscript is somewhat simple, and the theoretical problem addressed in it could have been identified through other means (for example through use of posterior predictive checking for model validation), and the actual behavioral data analyzed were interpreted as a null result (failure to reject that the behavioral stochasticity hypothesis), rather than actual identification of model misspecification. Thus, in my opinion, the jury is still out on whether this method could be used to identify a case of model misspecification that actually affects how individual differences are interpreted in a real cognitive task. Furthermore, the method requires considerable data for pretraining, well beyond what would be collected in a typical behavioral study, raising further questions about its applicability in problems of practical relevance. I expand on these primary concerns and raise several smaller points below.

A primary concern I have about this work is that it is unclear whether the method described could provide any advantage for real cognitive modeling problems beyond what is typically done to minimize the chance of model misspecification (in particular, posterior predictive checking). The toy problem examined in the manuscript is pretty extreme (two of the three synthetic agents are very far from what a human would do on the task, and the models deviate from one another to a degree that detecting the difference should not be difficult for any method). The issue posed in the toy data would easily be identified by following good modeling practices, which include using posterior predictive checking over summary measures to identify model insufficiencies, which in turn would call for the need for a broader set of models (See Wilson & Collins 2019). In this manuscript descriptive analyses are not performed (which, to me, feels a bit problematic for a paper that aims to improve cognitive modeling practices), however I think it is almost certain that the differences between the toy models would be evident by eye in standard summary measures of two-step task data. The primary question posed in the analysis of the empirical data is as to whether fit differences related IQ might reflect systematic differences in the model across individuals, but in this case application of the newly developed method provides little evidence for structural (model) differences. Thus, it remains unclear whether the method could identify model misspecification in real world data, and even more so whether it could reveal misspecification in situations where standard posterior predictive checking techniques would fall short. The rebuttal highlighted the better fit of the RNN on the empirical data as providing positive evidence for the ability of the method to identify model insufficiency, but I see this result as having limited epistemological value, given that there is no follow up to explore what the insufficiency actually was, or why accounting for it might be important. The authors list many of the points above as limitations in their discussion section, but in my opinion, they are relatively major ones.

The manuscript now mentions in the discussion that the newly developed methods should be seen as being just one tool in the larger toolkit of the computational cognitive modeler. However, one practical consideration here is that, since other existing tools such as simulation and descriptive analyses can be combined to (1) identify model insufficiency, (2) motivate specific model changes that can fix the problem, it is not exactly clear what the value added from the proposed method is.

One final practical limitation of the method is that it requires extensive pretraining (on >500 participants) in existing study, limiting its applicability for most use cases.

---

## [Referee Report · Reviewer #2 (Public review)]

SUMMARY:

In this manuscript, Ger and colleagues propose two complementary analytical methods aimed at quantifying the model misspecification and irreducible stochasticity in human choice behavior. The first method involves fitting recurrent neural networks (RNNs) and theoretical models to human choices and interpreting the better performance of RNNs as providing evidence of the misspecifications of theoretical models. The second method involves estimating the number of training iterations for which the fitted RNN achieves the best prediction of human choice behavior in a separate, validation data set, following an approach known as "early stopping". This number is then interpreted as a proxy for the amount of explainable variability in behavior, such that fewer iterations (earlier stopping) correspond to a higher amount of irreducible stochasticity in the data. The authors validate the two methods using simulations of choice behavior in a two-stage task, where the simulated behavior is generated by different known models. Finally, the authors use their approach in a real data set of human choices in the two-stage task, concluding that low-IQ subjects exhibit greater levels of stochasticity than high-IQ subjects.

STRENGTHS:

The manuscript explores an extremely important topic to scientists interested in characterizing human decision-making. While it is generally acknowledged that any computational model of behavior will be limited in its ability to describe a particular data set, one should hope to understand whether these limitations arise due to model misspecification or due to irreducible stochasticity in the data. Evidence for the former suggests that better models ought to exist; evidence for the latter suggests they might not.

To address this important topic, the authors elaborate carefully on the rationale of their proposed approach. They describe a variety of simulations -- for which the ground truth models and the amount of behavioral stochasticity are known -- to validate their approaches. This enables the reader to understand the benefits (and limitations) of these approaches when applied to the two-stage task, a task paradigm commonly used in the field. Through a set of convincing analyses, the authors demonstrate that their approach is capable of identifying situations where an alternative, untested computational model can outperform the set of tested models, before applying these techniques to a realistic data set.

WEAKNESSES:

The most significant weakness is that the paper rests on the implicit assumption that the fitted RNNs explain as much variance as possible, an assumption that is likely incorrect and which can result in incorrect conclusions. While in low-dimensional tasks RNNs can predict behavior as well as the data-generating models, this is not always the case, and the paper itself illustrates (in Figure 3) several cases where the fitted RNNs fall short of the ground-truth model. In such cases, we cannot conclude that a subject exhibiting a relatively poor RNN fit necessarily has a relatively high degree of behavioral stochasticity. Instead, it is at least conceivable that this subject's behavior is generated precisely (i.e., with low noise) by an alternative model that is pooly fit by an RNN -- e.g., a model with long-term sequential dependencies, which RNNs are known to have difficulties in capturing.

These situations could lead to incorrect conclusions for both of the proposed methods. First, the model mis-specification analysis might show equal predictive performance for a particular theoretical model and for the RNN. While a scientist might be inclined to conclude that the theoretical model explains the maximum amount of explainable variance and therefore that no better model should exist, the scenario in the previous paragraph suggests that a superior model might nonetheless exist. Second, in the early-stopping analysis, a particular subject may achieve optimal validation performance with fewer epochs than another, leading the scientist to conclude that this subject exhibits higher behavioral noise. However, as before, this could again result from the fact that this subject's behavior is produced with little noise by a different model. The possibility of such scenarios does not mean that such scenarios are common, and the conclusions drawn in the paper are likely appropriate for the particular examples analyzed. However, it is much less obvious that the RNNs will provide optimal fits in other types of tasks, particularly those with more complex rules and long-term sequential dependencies, and in such scenarios, an ill-advised scientist might end up drawing incorrect conclusions from the application of the proposed approaches. The authors acknowledge this limitation in their discussion, but it remains a significant caveat that readers should be aware of when using the technique proposed.

In addition to this general limitation, the relationship between the number of optimal epochs and behavioral stochasticity may not hold for every task and every subject. For example, Figure 4 highlights the relationship between the optimal epochs and agent noise. Yet, it is nonetheless possible that the optimal epoch is influenced by model parameters other than inverse temperature (e.g., hyperparameters such as learning rate, etc). This could again lead to invalid conclusions, such as concluding that low-IQ is associated with optimal epoch when an alternative account might be that low-IQ is associated with low learning rate, which in turn is associated with optimal epoch. Additional factors such as the deep double-descent (Nakkiran et al., ICLR 2020) can also influence the optimal epoch value as computed by the authors. These concerns are partially addressed by the authors in the revised manuscript, where they show that the number of optimal epochs is primarily sensitive to the amount of true underlying noise, assuming the number of trials and network size are constant. The authors also acknowledge, in the discussion section, that many factors can affect the number of optimal epochs, and that inferring behavioral stochasticity from this number should be done with caution.

APPRAISAL AND DISCUSSION:

Overall, the authors propose a novel method that aims to solve an important problem, but since the evidence provided refers to a single task and to a single dataset, it is not clear that the method would be appropriate in general settings. In the future, it would be beneficial to test the proposed approach in a broader setting, including simulations of different tasks, different model classes, and different model parameters. Nonetheless, even without such additional work, the proposed methods are likely to be used by cognitive scientists and neuroscientists interested in assessing the quality and limits of their behavioral models.

---

## [Author Response]

The following is the authors’ response to the previous reviews.

**eLife assessment**
In this study, Ger and colleagues present a valuable new technique that uses recurrent neural networks to distinguish between model misspecification and behavioral stochasticity when interpreting cognitivebehavioral model fits. Evidence for the usefulness of this technique, which is currently based primarily on a relatively simple toy problem, is considered incomplete but could be improved via comparisons to existing approaches and/or applications to other problems. This technique addresses a long-standing problem that is likely to be of interest to researchers pushing the limits of cognitive computational modeling.
**Public Reviews:**

**Reviewer #1 (Public Review):**
Summary:Ger and colleagues address an issue that often impedes computational modeling: the inherent ambiguity between stochasticity in behavior and structural mismatch between the assumed and true model. They propose a solution to use RNNs to estimate the ceiling on explainable variation within a behavioral dataset. With this information in hand, it is possible to determine the extent to which "worse fits" result from behavioral stochasticity versus failures of the cognitive model to capture nuances in behavior (model misspecification). The authors demonstrate the efficacy of the approach in a synthetic toy problem and then use the method to show that poorer model fits to 2-step data in participants with low IQ are actually due to an increase in inherent stochasticity, rather than systemic mismatch between model and behavior.Strengths:Overall I found the ideas conveyed in the paper interesting and the paper to be extremely clear and wellwritten. The method itself is clever and intuitive and I believe it could be useful in certain circumstances, particularly ones where the sources of structure in behavioral data are unknown. In general, the support for the method is clear and compelling. The flexibility of the method also means that it can be applied to different types of behavioral data - without any hypotheses about the exact behavioral features that might be present in a given task.

Thank you for taking the time to review our work and for the positive remarks regarding the manuscript. Below is a point-by-point response to the concerns raised.

Weaknesses:That said, I have some concerns with the manuscript in its current form, largely related to the applicability of the proposed methods for problems of importance in computational cognitive neuroscience. This concern stems from the fact that the toy problem explored in the manuscript is somewhat simple, and the theoretical problem addressed in it could have been identified through other means (for example through the use of posterior predictive checking for model validation), and the actual behavioral data analyzed were interpreted as a null result (failure to reject that the behavioral stochasticity hypothesis), rather than actual identification of model-misspecification. I expand on these primary concerns and raise several smaller points below.A primary question I have about this work is whether the method described would actually provide any advantage for real cognitive modeling problems beyond what is typically done to minimize the chance of model misspecification (in particular, post-predictive checking). The toy problem examined in the manuscript is pretty extreme (two of the three synthetic agents are very far from what a human would do on the task, and the models deviate from one another to a degree that detecting the difference should not be difficult for any method). The issue posed in the toy data would easily be identified by following good modeling practices, which include using posterior predictive checking over summary measures to identify model insufficiencies, which in turn would call for the need for a broader set of models (See Wilson & Collins 2019). Thus, I am left wondering whether this method could actually identify model misspecification in real world data, particularly in situations where standard posterior predictive checking would fall short. The conclusions from the main empirical data set rest largely on a null result, and the utility of a method for detecting model misspecification seems like it should depend on its ability to detect its presence, not just its absence, in real data.Beyond the question of its advantage above and beyond data- and hypothesis-informed methods for identifying model misspecification, I am also concerned that if the method does identify a modelinsufficiency, then you still would need to use these other methods in order to understand what aspect of behavior deviated from model predictions in order to design a better model. In general, it seems that the authors should be clear that this is a tool that might be helpful in some situations, but that it will need to be used in combination with other well-described modeling techniques (posterior predictive checking for model validation and guiding cognitive model extensions to capture unexplained features of the data). A general stylistic concern I have with this manuscript is that it presents and characterizes a new tool to help with cognitive computational modeling, but it does not really adhere to best modeling practices (see Collins & Wilson, eLife), which involve looking at data to identify core behavioral features and simulating data from best-fitting models to confirm that these features are reproduced. One could take away from this paper that you would be better off fitting a neural network to your behavioral data rather than carefully comparing the predictions of your cognitive model to your actual data, but I think that would be a highly misleading takeaway since summary measures of behavior would just as easily have diagnosed the model misspecification in the toy problem, and have the added advantage that they provide information about which cognitive processes are missing in such cases.As a more minor point, it is also worth noting that this method could not distinguish behavioral stochasticity from the deterministic structure that is not repeated across training/test sets (for example, because a specific sequence is present in the training set but not the test set). This should be included in the discussion of method limitations. It was also not entirely clear to me whether the method could be applied to real behavioral data without extensive pretraining (on >500 participants) which would certainly limit its applicability for standard cases.The authors focus on model misspecification, but in reality, all of our models are misspecified to some degree since the true process-generating behavior almost certainly deviates from our simple models (ie. as George Box is frequently quoted, "all models are wrong, but some of them are useful"). It would be useful to have some more nuanced discussion of situations in which misspecification is and is not problematic.

We thank the reviewer for these comments and have made changes to the manuscript to better describe these limitations. We agree with the reviewer and accept that fitting a neural network is by no means a substitute for careful and dedicated cognitive modeling. Cognitive modeling is aimed at describing the latent processes that are assumed to generate the observed data, and we agree that careful description of the data-generating mechanisms, including posterior predictive checks, is always required. However, even a well-defined cognitive model might still have little predictive accuracy, and it is difficult to know how much resources should be put into trying to test and develop new cognitive models to describe the data. We argue that RNN can lead to some insights regarding this question, and highlight the following limitations that were mentioned by the review:

First, we accept that it is important to provide positive evidence for the existence of model misspecification. In that sense, a result where the network shows dramatic improvement over the best-fitting theoretical model is easier to interpret compared to when the network shows no (or very little) improvement in predictive accuracy. This is because there is always an option that the network, for some reason, was not flexible enough to learn the data-generating model, or because the data-generating mechanism has changed from training to test. We have now added this more clearly in the limitation section. However, when it comes to our empirical results, we would like to emphasize that the network did in fact improve the predictive accuracy for all participants. The result shows support in favor of a "null" hypothesis in the sense that we seem to find evidence that the change in predictive accuracy between the theoretical model and RNN is not systematic across levels of IQ. This allows us to quantify evidence (use Bayesian statistics) for no systematic model misspecification as a function of IQ. While it is always possible that a different model might systematically improve the predictive accuracy of low vs high IQ individuals' data, this seems less likely given the flexibility of the current results.

Second, we agree that our current study only applies to the RL models that we tested. In the context of RL, we have used a well-established and frequently applied paradigm and models. We emphasize in the discussion that simulations are required to further validate other uses for this method with other paradigms.

Third, we also accept that posterior predictive checks should always be capitalized when possible, which is now emphasized in the discussion. However, we note that these are not always easy to interpret in a meaningful way and may not always provide details regarding model insufficiencies as described by the reviewer. It is very hard to determine what should be considered as a good prediction and since the generative model is always unknown, sometimes very low predictive accuracy can still be at the peak of possible model performance. This is because the data might be generated from a very noisy process, capping the possible predictive accuracy at a very low point. However, when strictly using theoretical modeling, it is very hard to determine what predictive accuracy to expect. Also, predictive checks are not always easy to interpret visually or otherwise. For example, in two-armed bandit tasks where there are only two actions, the prediction of choices is easier to understand in our opinion when described using a confusion matrix that summarizes the model's ability to predict the empirical behavior (which becomes similar to the predictive estimation we describe in eq 22).

Finally, this approach indeed requires a large dataset, with at least three sessions for each participant (training, validation, and test). Further studies might shed more light on the use of optimal epochs as a proxy for noise/complexity that can be used with less data (i.e., training and validation, without a test set).

Please see our changes at the end of this document.

**Reviewer #2 (Public Review):**
SUMMARY:In this manuscript, Ger and colleagues propose two complementary analytical methods aimed at quantifying the model misspecification and irreducible stochasticity in human choice behavior. The first method involves fitting recurrent neural networks (RNNs) and theoretical models to human choices and interpreting the better performance of RNNs as providing evidence of the misspecifications of theoretical models. The second method involves estimating the number of training iterations for which the fitted RNN achieves the best prediction of human choice behavior in a separate, validation data set, following an approach known as "early stopping". This number is then interpreted as a proxy for the amount of explainable variability in behavior, such that fewer iterations (earlier stopping) correspond to a higher amount of irreducible stochasticity in the data. The authors validate the two methods using simulations of choice behavior in a two-stage task, where the simulated behavior is generated by different known models. Finally, the authors use their approach in a real data set of human choices in the two-stage task, concluding that low-IQ subjects exhibit greater levels of stochasticity than high-IQ subjects.STRENGTHS:The manuscript explores an extremely important topic to scientists interested in characterizing human decision-making. While it is generally acknowledged that any computational model of behavior will be limited in its ability to describe a particular data set, one should hope to understand whether these limitations arise due to model misspecification or due to irreducible stochasticity in the data. Evidence for the former suggests that better models ought to exist; evidence for the latter suggests they might not.To address this important topic, the authors elaborate carefully on the rationale of their proposed approach. They describe a variety of simulations - for which the ground truth models and the amount of behavioral stochasticity are known - to validate their approaches. This enables the reader to understand the benefits (and limitations) of these approaches when applied to the two-stage task, a task paradigm commonly used in the field. Through a set of convincing analyses, the authors demonstrate that their approach is capable of identifying situations where an alternative, untested computational model can outperform the set of tested models, before applying these techniques to a realistic data set.

Thank you for reviewing our work and for the positive tone. Please find below a point-by-point response to the concerns you have raised.

WEAKNESSES:The most significant weakness is that the paper rests on the implicit assumption that the fitted RNNs explain as much variance as possible, an assumption that is likely incorrect and which can result in incorrect conclusions. While in low-dimensional tasks RNNs can predict behavior as well as the data-generating models, this is not *always* the case, and the paper itself illustrates (in Figure 3) several cases where the fitted RNNs fall short of the ground-truth model. In such cases, we cannot conclude that a subject exhibiting a relatively poor RNN fit necessarily has a relatively high degree of behavioral stochasticity. Instead, it is at least conceivable that this subject's behavior is generated precisely (i.e., with low noise) by an alternative model that is poorly fit by an RNN - e.g., a model with long-term sequential dependencies, which RNNs are known to have difficulties in capturing.These situations could lead to incorrect conclusions for both of the proposed methods. First, the model misspecification analysis might show equal predictive performance for a particular theoretical model and for the RNN. While a scientist might be inclined to conclude that the theoretical model explains the maximum amount of explainable variance and therefore that no better model should exist, the scenario in the previous paragraph suggests that a superior model might nonetheless exist. Second, in the earlystopping analysis, a particular subject may achieve optimal validation performance with fewer epochs than another, leading the scientist to conclude that this subject exhibits higher behavioral noise. However, as before, this could again result from the fact that this subject's behavior is produced with little noise by a different model. Admittedly, the existence of such scenarios *in principle* does not mean that such scenarios are common, and the conclusions drawn in the paper are likely appropriate for the particular examples analyzed. However, it is much less obvious that the RNNs will provide optimal fits in other types of tasks, particularly those with more complex rules and long-term sequential dependencies, and in such scenarios, an ill-advised scientist might end up drawing incorrect conclusions from the application of the proposed approaches.

Yes, we understand and agree. A negative result where RNN is unable to overcome the best fitting theoretical model would always leave room for doubt regarding the fact that a different approach might yield better results. In contrast, a dramatic improvement in predictive accuracy for RNN is easier to interpret since it implies that the theoretical model can be improved. We have made an effort to make this issue clear and more articulated in the discussion. We specifically and directly mention in the discussion that “Equating RNN performance with the generative model should be avoided”.

However, we would like to note that our empirical results provided a somewhat more nuanced scenario where we found that the RNN generally improved the predictive accuracy of most participants. Importantly, this improvement was found to be equal across participants with no systematic benefits for low vs high IQ participants. We understand that there is always the possibility that another model would show a systematic benefit for low vs. high IQ participants, however, we suggest that this is less likely given the current evidence. We have made an effort to clearly note these issues in the discussion.

In addition to this general limitation, the paper also makes a few additional claims that are not fully supported by the provided evidence. For example, Figure 4 highlights the relationship between the optimal epochs and agent noise. Yet, it is nonetheless possible that the optimal epoch is influenced by model parameters other than inverse temperature (e.g., learning rate). This could again lead to invalid conclusions, such as concluding that low-IQ is associated with optimal epoch when an alternative account might be that low-IQ is associated with low learning rate, which in turn is associated with optimal epoch. Yet additional factors such as the deep double-descent (Nakkiran et al., ICLR 2020) can also influence the optimal epoch value as computed by the authors.An additional issue is that Figure 4 reports an association between optimal epoch and noise, but noise is normalized by the true minimal/maximal inverse-temperature of hybrid agents (Eq. 23). It is thus possible that the relationship does not hold for more extreme values of inverse-temperature such as beta=0 (extremely noisy behavior) or beta=inf (deterministic behavior), two important special cases that should be incorporated in the current study. Finally, even taking the association in Figure 4 at face value, there are potential issues with inferring noise from the optimal epoch when their correlation is only r~=0.7. As shown in the figures, upon finding a very low optimal epoch for a particular subject, one might be compelled to infer high amounts of noise, even though several agents may exhibit a low optimal epoch despite having very little noise.

Thank you for these comments. Indeed, there is much we do not yet fully understand about the factors that influence optimal epochs. Currently, it is clear to us that the number of optimal epochs is influenced by a variety of factors, including network size, the data size, and other cognitive parameters, such as the learning rate. We hope that our work serves as a proof-of-concept, suggesting that, in certain scenarios, the number of epochs can be utilized as an empirical estimate. Moreover, we maintain that, at least within the context of the current paradigm, the number of optimal epochs is primarily sensitive to the amount of true underlying noise, assuming the number of trials and network size are constant. We are therefore hopeful that this proofof-concept will encourage research that will further examine the factors that influence the optimal epochs in different behavioral paradigms.

To address the reviewer's justified concerns, we have made several amendments to the manuscript. First, we added an additional version of Figure 4 in the Supplementary Information material, where the noise parameter values are not scaled. We hope this adjustment clarifies that the parameters were tested across a broad spectrum of values (e.g., 0 to 10 for the hybrid model), spanning the two extremes of complete randomness and high determinism. Second, we included a linear regression analysis showing the association of all model parameters (including noise) with the optimal number of epochs. As anticipated by the reviewer, the learning rate was also found to be associated with the number of optimal epochs. Nonetheless, the noise parameter appears to maintain the most substantial association with the number of optimal epochs. We have also added a specific mentioning of these associations in the discussion, to inform readers that the association between the number of optimal epochs and model parameters should be examined using simulation for other paradigms/models. Lastly, we acknowledge in the discussion that the findings regarding the association between the number of optimal epochs and noise warrant further investigation, considering other factors that might influence the determination of the optimal epoch point and the fact that the correlation with noise is strong, but not perfect (in the range of 0.7).

The discussion now includes the following:

“Several limitations should be considered in our proposed approach. First, fitting a data-driven neural network is evidently not enough to produce a comprehensive theoretical description of the data generation mechanisms. Currently, best practices for cognitive modeling (Wilson and Collins, 2019) require identifying under what conditions the model struggles to predict the data (e.g., using posterior predictive checks), and describing a different theoretical model that could account for these disadvantages in prediction. However, identifying conditions where the model shortcomings in predictive accuracy are due to model misspecifications rather than noisier behavior is a challenging task. We propose leveraging data-driven RNNs as a supplementary tool, particularly when they significantly outperform existing theoretical models, followed by refined theoretical modeling to provide insights into what processes were mis-specified in the initial modeling effort.

Second, although we observed a robust association between the optimal number of epochs and true noise across varying network sizes and dataset sizes (see Figure S2), additional factors such as network architecture and other model parameters (e.g., learning rate, see .Figure S7}) might influence this estimation. Further research is required to allow us to better understand how and why different factors change the number of optimal epochs for a given dataset before it can be applied with confidence to empirical investigations.

Third, the empirical dataset used in our study consisted of data collected from human participants at a single time point, serving as the training set for our RNN. The test set data, collected with a time interval of approximately $\sim6$ and $\sim18$ months, introduced the possibility of changes in participants' decision-making strategies over time. In our analysis, we neglected any possible changes in participants' decision-making strategies during that time, changes that may lead to poorer generalization performance of our approach. Thus, further studies are needed to eliminate such possible explanations.

Fourth, our simulations, albeit illustrative, were confined to known models, necessitating in-silico validation before extrapolating the efficacy of our approach to other model classes and tasks. Our aim was to showcase the potential benefits of using a data-driven approach, particularly when faced with unknown models. However, whether RNNs will provide optimal fits for tasks with more complex rules and long-term sequential dependencies remains uncertain.

Finally, while positive outcomes where RNNs surpass theoretical models can prompt insightful model refinement, caution is warranted in directly equating RNN performance with that of the generative model, as seen in our simulations (e.g., Figure 3). We highlight that our empirical findings depict a more complex scenario, wherein the RNN enhanced the predictive accuracy for all participants uniformly. Notably, we also provide evidence supporting a null effect among individuals, with no consistent difference in RNN improvement over the theoretical model based on IQ. Although it remains conceivable that a different datadriven model could systematically heighten the predictive accuracy for individuals with lower IQs in this task, such a possibility seems less probable in light of the current findings.”

**Reviewer #1 (Recommendations For The Authors):**
Minor comments:Is the t that gets fed as input to RNN just timestep?

t = last transition type (rare/common). not timestep

Line 378: what does "optimal epochs" mean here?

The number of optimal training epochs that minimize both underfitting and overfitting (define in the line ~300)

Line 443: I don't think "identical" is the right word here - surely the authors just mean that there is not an obvious systematic difference in the distributions.

Fixed

I was expecting to see ~500 points in Figure 7a, but there seem to be only 50... why weren't all datasets with at least 2 sessions used for this analysis?

We used the ~500 subjects (only 2 datasets) to pre-train the RNN, and then fine-tuned the pre-trained RNN on the other 54 subjects that have 3 datasets. The correlation of IQ and optimal epoch also hold for the 500 subjects as shown below.

**Reviewer #2 (Recommendations For The Authors):**
Figure 3b: despite spending a long time trying to understand the meaning of each cell of the confusion matrix, I'm still unsure what they represent. Would be great if you could spell out the meaning of each cell individually, at least for the first matrix in the paper.

We added a clarification to the Figure caption.

Figure 5: Why didn't the authors show this exact scenario using simulated data? It would be much easier to understand the predictions of this figure if they had been demonstrated in simulated data, such as individuals with different amounts of behavioral noise or different levels of model misspecifications.

In Figure 5 the x-axis represents IQ. Replacing the x-axis with true noise would make what we present now as Figure 4. We have made an effort to emphasize the meaning of the axes in the caption.

Line 195 ("...in the action selection. Where"). Typo? No period is needed before "where".

Fixed

Line 213 ("K dominated-hand model"). I was intrigued by this model, but wasn't sure whether it has been used previously in the literature, or whether this is the first time it has been proposed.

This is the first time that we know of that this model is used.

Line 345 ("This suggests that RNN is flexible enough to approximate a wide range of different behavioral models"): Worth explaining why (i.e., because the GRUs are able to capture dependencies across longer delays than a k-order Logistic Regression model).Line 356 ("We were interested to test"): Suggestion: "We were interested in testing".

Fixed

Line 389 ("However, as long as the number of observations and the size of the network is the same between two datasets, the number of optimal epochs can be used to estimate whether the dataset of one participant is noisier compared with a second dataset."): This is an important claim that should ideally be demonstrated directly. The paper only illustrates this effect through a correlation and a scatter plot, where higher noise tends to predict a lower optimal epoch. However, is the claim here that, in some circumstances, optimal epoch can be used to *deterministically* estimate noise? If so, this would be a strong result and should ideally be included in the paper.

We have now omitted this sentenced and toned down our claims, suggesting that while we did find a strong association between noise and optimal epochs, future research is required to established to what extent this could be differentiated from other factors (i.e., network size, amount of observations).